# B-cosification: Transforming Deep Neural Networks to be Inherently Interpretable

**Shreyash Arya**[*,1,2], **Sukrut Rao**[*,1,2], **Moritz Böhle**[*,†,1,3], **Bernt Schiele**[1]

[1]Max Planck Institute for Informatics, Saarland Informatics Campus, Saarbrücken, Germany
[2]RTG Neuroexplicit Models of Language, Vision, and Action, Saarbrücken, Germany
[3] Kyutai, Paris, France
{sarya,sukrut.rao,schiele}@mpi-inf.mpg.de   moritz@kyutai.org
[*]Equal contribution   [†] Work done while at MPI Informatics

## Abstract

B-cos Networks have been shown to be effective for obtaining highly human interpretable explanations of model decisions by architecturally enforcing stronger alignment between inputs and weight. B-cos variants of convolutional networks (CNNs) and vision transformers (ViTs), which primarily replace linear layers with B-cos transformations, perform competitively to their respective standard variants while also yielding explanations that are faithful by design. However, it has so far been necessary to train these models from scratch, which is increasingly infeasible in the era of large, pre-trained foundation models. In this work, inspired by the architectural similarities in standard DNNs and B-cos networks, we propose 'B-cosification', a novel approach to *transform* existing pre-trained models to become inherently interpretable. We perform a thorough study of design choices to perform this conversion, both for convolutional neural networks and vision transformers. We find that B-cosification can yield models that are on par with B-cos models trained from scratch in terms of interpretability, while often outperforming them in terms of classification performance at a fraction of the training cost. Subsequently, we apply B-cosification to a pretrained CLIP model, and show that, even with limited data and compute cost, we obtain a B-cosified version that is highly interpretable and competitive on zero shot performance across a variety of datasets. We release our code and pre-trained model weights at https://github.com/shrebox/B-cosification.

## 1 Introduction

Despite their strong performance on a variety of tasks, understanding decisions of deep neural networks (DNNs) remains challenging. Explanation methods, such as feature attributions [46, 48, 53, 5], have been proposed in an attempt to explain such decisions *post-hoc*, but have often found to be unfaithful to the model being explained [2, 3, 40, 60].

Inherently interpretable Deep Neural Network (DNN) models have recently gained popularity. In contrast to the common approach of explaining existing DNNs in a *post-hoc* fashion, these models typically feature certain architectural constraints that allow for extracting human-interpretable, model-faithful simplifications of the models' computations *by design*; examples of this include prototype-based [14, 17, 32], dynamic linear [9, 10], or concept-bottleneck models [28, 58, 33, 42]. However, given those architectural changes, this comes at a price: specifically, the models need to be trained from scratch, which—especially in the case of large foundation models, which are increasingly popular—can cost millions of dollars.

To mitigate this, in this work, we explore a novel approach of *fine-tuning DNNs for inherent interpretability* and propose to 'B-cosify' existing DNNs. Specifically, we investigate whether pre-trained DNNs can simply be efficiently fine-tuned to obtain a similar degree of interpretability as

38th Conference on Neural Information Processing Systems (NeurIPS 2024).

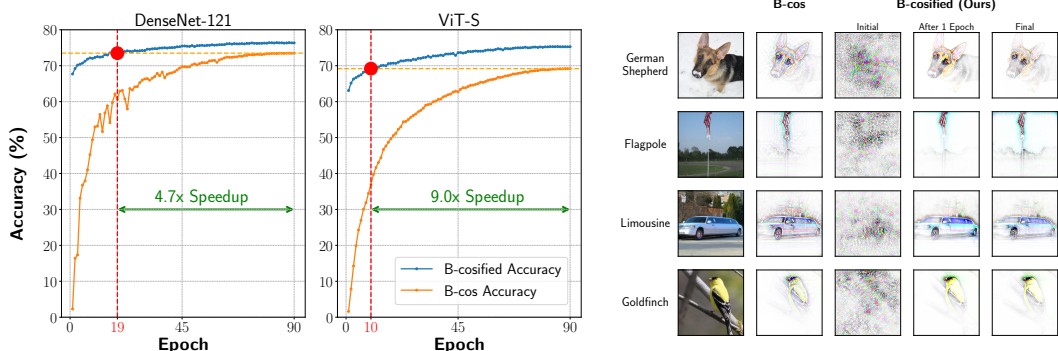

Fig. 1: **B-cosification: Obtaining inherently interpretable models with competitive accuracy at low cost.** *Left:* Accuracy progression over epochs for a DenseNet-121 and a ViT-S, comparing B-cosified (blue) and B-cos (orange) training curves. B-cosified models achieve equivalent accuracy with a substantial reduction in training time, yielding 4.7x speedup for DenseNet-121 and 9.0x speedup for ViT-S. *Right:* Qualitative comparison of explanations for various images for B-cos [11] and our B-cosified models at various stages of training. Specifically, we show the dynamic linear mappings $\mathbf{W}(\mathbf{x})$ computed by the models in color as in [10, 11]; note that by formulating conventional models ('initial' in the plot) as a specific version of B-cos models, we are able to visualise the corresponding explanations in color too, see Sec. 3.2.1 for further details. We find that after only one epoch of training, the B-cosified models exhibit similar explanations as B-cos models.

the recently proposed B-cos Networks [10, 11]. In contrast to the original B-cos Networks, which leverage existing *architectures* to obtain performant and interpretable models, we investigate whether we can additionally leverage the existing pre-trained *weights*, thus aiming to take advantage of the significant amount of resources that have been invested in training existing models. As a result, we hope to make inherently interpretable models more easily accessible to the community.

To do so, we first conduct a detailed analysis of how B-cos DNNs differ from their conventional counterparts. Interestingly, we find that many existing models can be converted into *functionally equivalent* B-cos models by a small set of targeted implementational modifications (Tab. 1). To increase the interpretability of the models, we then increase the 'alignment pressure' [10] via the parameter B of the B-cos transformations and fine-tune the models on their respective tasks, which leads to significantly more interpretable explanations (Fig. 2).

On supervised settings, we find that B-cosified models often outperform both conventional and B-cos DNNs at a fraction of the full training cost (Fig. 1, left), whilst exhibiting a similar degree of interpretability as the original B-cos DNNs (Fig. 1, right). We further apply B-cosification to a pre-trained CLIP model [39], a large foundation vision-language model (VLM), and show that despite using comparatively limited data and compute cost, B-cosified CLIP models yield highly interpretable explanations whilst being competitive on zero-shot performance across a variety of downstream datasets.

Our work thus opens a new perspective on how to design inherently interpretable models in a cost-effective manner. Importantly, on the one hand it highlights that conventional models might be closer to inherently interpretable models than previously understood. On the other hand, it highlights the benefits of designing inherently interpretable models via minor architectural modifications, such as e.g. the B-cos DNNs, as this can allow for leveraging the large array of existing, pre-trained DNNs.

In summary, our contributions are:

- We propose *B-cosification*, a novel technique to 'fine-tune for interpretability', that addresses the problem of high training cost associated with obtaining inherently intepretable models such as B-cos DNNs. Our B-cosified DNNs are highly interpretable while often outperforming both standard and B-cos DNNs.

- We thoroughly study different design choices to find an optimal strategy for B-cosification.

- We apply B-cosification to supervised image classifiers on ImageNet [16], including both CNNs and ViTs, and show that the B-cosified variants perform on par on interpretability metrics while often outperforming in terms of accuracy. Overall, we find that B-cosifying a pre-trained black box DNN to be superior on both metrics as compared to training a B-cos DNN from scratch, while being computationally significantly cheaper.

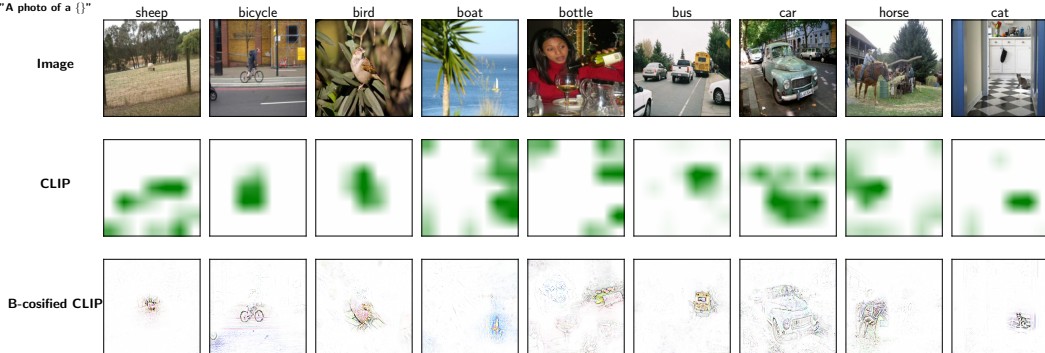

Fig. 2: **B-cosified CLIP Models.** After B-cosifying a CLIP model and fine-tuning it according to our proposed B-cosification scheme, see Sec. 3.2, we find that it is possible to endow the model with the same level of inherent interpretability as the B-cos models proposed in [11], whilst maintaining CLIP's zeroshot ability (see Fig. 5). The resulting linear summaries of the models ($\mathbf{W}(\mathbf{x})$) can be visualised in color (row 3) and provide significantly more detail than GradCAM explanations (row 2), which are often used to explain conventional CLIP models.

- We extend B-cosification to CLIP, a foundation VLM, and show that B-cosified CLIP remains highly competitive on zero-shot performance across a variety of downstream datasets, while also yielding similar interpretability benefits as B-cos models.

## 2 Related Work

**Explanation Methods.** Post-hoc attributions [46, 43, 53, 48, 5, 56] have popularly been used to understand the decisions of trained DNNs, but have often been shown to be unfaithful to the model being explained [2, 3, 60, 40]. Inherently interpretable models [14, 28, 10], in contrast, incorporate architectural changes to the model and can yield explanations that are interpretable and faithful to the model by design. However, such models need to be trained from scratch, which imposes a significant additional cost. In this work, we explore fine-tuning for interpretability, and propose a method to transform existing black-box DNNs to inherently interpretable B-cos DNNs, bringing together the best of both worlds.

**Attribution Priors** [38, 37, 27, 50, 4] have often been used to train or fine-tune models to have explanations with desirable properties, such as inducing smoother explanations [27], consistent explanations [37, 38], or to guide models to be 'right for the right reasons' [44, 20, 41, 34]. Similar to such work, we fine-tune black-box DNNs for interpretability, but in contrast, we only make *architectural* modifications to transform the DNNs to B-cos DNNs, and do not use any additional constraints on the explanations themselves while training.

**CLIP Interpretability and Localization.** Post-hoc attribution methods [46, 36, 13, 7] have also been used to explain VLMs such as CLIP [39], however, as with supervised DNNs, their faithfulness to the model is not guaranteed and the explanations are often coarse-grained and not very human interpretable. While inherently interpretable architectures could address this, the high costs of training such large models from scratch makes their use unappealing. In this work, we bridge the gap by instead fine-tuning from pre-trained black-box CLIP models to inherently interpretable B-cosified CLIP variants, and find that the B-cosification process is effective in yielding performant and interpretable models. A separate line of work [8] involves improving localizability of VLMs, and is orthogonal to our work since our goal is to obtain explanations that are faithful to the model.

**Learning mappings between model features.** Recent work [30, 33] has explored using simple linear transforms to map features between models, and in particular also mapping features from arbitrary models to CLIP's representation space. In the context of our work, such methods can be used to map a supervised B-cos feature extractor to CLIP using a linear transform, to obtain an inherently interpretable DNN that can mimic CLIP. In our evaluation, we compare with such an approach, and find that our approach of architecturally transforming the full model and fine-tuning for interpretability yields improved zero shot performance.

## 3 From conventional to B-cos models

In the following, we describe the process of fine-tuning standard black box DNNs into inherently interpretable B-cos DNNs. In Sec. 3.1, we first introduce the B-cos models and enumerate the key ways in which they differ from standard models. In Sec. 3.2, we then perform a detailed study on strategies to bridge each of these differences for effective B-cosification.

### 3.1 B-cos Models: Background

Many common DNNs consist of a series of blocks of linear layers followed by non-linear ReLU activations [31], and are thus piece-wise linear functions[1]: i.e., for every input $\mathbf{x}$, they effectively compute a linear transformation of that input: $\mathbf{y}(\mathbf{x}) = \mathbf{W}(\mathbf{x})\mathbf{x} + \mathbf{b}(\mathbf{x})$, cf. [52]. In [9, 11], models of this kind have been called 'dynamic linear', a naming convention that we adopt in this paper.

Interestingly, for piece-wise linear models, $\mathbf{W}(\mathbf{x})$ is given by the models' gradient with respect to $\mathbf{x}$ [52]—except for the input-dependent bias $\mathbf{b}(\mathbf{x})$, the gradient thus constitutes an exact summary of the models' computations. This linear mapping $\mathbf{W}(\mathbf{x})$ is unfortunately typically not easily interpretable, and many techniques have been proposed to derive qualitatively more convincing explanations [46, 56]. These, however, have been shown to often not faithfully reflect the underlying model [1, 40].

Further, if the models employ bias terms, $\mathbf{W}(\mathbf{x})$ does not yield a *complete* explanation [52], i.e. $\mathbf{y}(\mathbf{x}) \neq \mathbf{W}(\mathbf{x})\mathbf{x}$. Integrating bias terms as proposed by [52] yields a set of importance attribution maps, summarizing which requires carefully selecting a post-processing function with inherent tradeoffs. Even when not using bias terms [23], however, the resulting matrices $\mathbf{W}(\mathbf{x})$ are often not easily human interpretable, and the resulting models can suffer from significant drops in performance.

To address this, [10, 11] propose to architecturally modify the DNNs to introduce additional *alignment pressure* during model optimisation. For this, they replace the ubiquitously used linear transformation by the B-cos transformation, which dynamically scales the output of the linear transformations:

**B-cos transformation:** $\qquad f_{\text{B-cos}}(\mathbf{x}; \mathbf{w}) = \left( |\cos(\mathbf{x}, \mathbf{w})|^{B-1} \times \widehat{\mathbf{w}} \right)^T \mathbf{x} = \mathbf{w}^T(\mathbf{x})\mathbf{x} ,$ $\qquad$ (1)

with $B$ a hyperparameter, $\cos$ the cosine similarity between $\mathbf{x}$ and the weights $\mathbf{w}$, and $\widehat{\mathbf{w}} = \mathbf{w}/\|\mathbf{w}\|$.

Like piece-wise linear models, B-cos models are dynamic linear and thus accurately summarised by a single linear transformation $\mathbf{W}(\mathbf{x})$ s.t. $\mathbf{y}(\mathbf{x}) = \mathbf{W}(\mathbf{x})\mathbf{x}$; as B-cos models do not employ bias terms, this model summary is complete. Crucially, it has been shown that with $B > 1$, the matrix $\mathbf{W}(\mathbf{x})$ aligns with task-relevant input patterns, making it more easily human interpretable (e.g. Fig. 1, right).

Importantly, as the B-cos transformation can serve as a drop-in replacement for linear transformations at every layer of a DNN, it is possible [10, 11] to leverage *existing DNN architectures* and the resulting B-cos models obtain similar classification accuracies as their conventional counterparts (Tab. 4, cols. 'pretrained' and 'B-cos').

Extending this, we investigate if it is possible to leverage *existing DNN weights*—i.e., our goal is to fine-tune existing models to be similarly interpretable as B-cos models, whilst not requiring to train them from scratch. However, despite the architectural similarities between B-cos and conventional models, there are multiple key differences that make transforming pre-trained models into B-cos models non-trivial: e.g., apart from replacing linear transformations with the B-cos transformation and not employing biases, B-cos models are trained on image representations with 6 color channels as $[r, g, b, 1-r, 1-g, 1-b]$ to be able to visualise the model-inherent linear summaries $\mathbf{W}(\mathbf{x})$ in color, whereas conventional models use 3 channels (see also Tab. 1). In the next section, we show how to overcome these differences and convert existing models into functionally equivalent B-cos models.

### 3.2 B-cosification of Deep Neural Networks

We analysed the differences between B-cos models and their conventional counterparts in detail and compiled the results in Tab. 1. In this section, we discuss one by one how to bridge these differences. In particular, we show that a conventional model can be framed as a *functionally equivalent* B-cos model as in [11] with $B = 1$, which additionally employs bias terms. Only upon modifying these two aspects, i.e. biases and $B$, does the model need to be fine-tuned to adapt the weights to those changes.

---

[1]As noted by [52], 'piece-wise linear' is actually a misnomer. As the models additionally employ biases, the resulting DNNs are in fact *piece-wise affine*. For simplicity, we maintain the common naming convention.

Table 1: **Overview.** To allow for comparing the models, we compiled the identified differences between the conventional models (**Standard**), their B-cosified version (**B-cosified**) and the original B-cos models (**B-cos**). For each design choice in the B-cosified models, we summarise the respective discussion in Sec. 3.2 (**reason**).

| Property | Standard | B-cos | B-cosified | reason |
|---|---|---|---|---|
| **Image Encoding** | 3 channels | 6 channels | 6 channels | $\rightarrow$ colored explanations |
| **Normalized Inputs** | yes | no | yes | $\rightarrow$ in-distribution (ID) |
| **Weights** | unnormalised | normalised | unnormalised | $\rightarrow$ equivalent and ID |
| **Activations** | ReLU | none | ReLU | $\rightarrow$ compatible and ID |
| **Biases** | yes | no | no | $\rightarrow$ complete explanations |
| $B$ **in B-cos** | 1 | 2 | 2 | $\rightarrow$ weight-input alignment |

### 3.2.1 Functionally Equivalent B-cos Models

**Input Encoding and Normalisation.** As mentioned in Sec. 3.1, B-cos models use input representations with six color channels $[r, g, b, 1-r, 1-g, 1-b]$ to be able to visualise the explanations in color, cf. [11]. However, most conventional DNNs (e.g. models from Torchvision [54], CLIP [39]) are applied to 3-channel inputs in which images are encoded via $[r, g, b]$. As a result, visualising the dynamic matrices $\mathbf{W}(\mathbf{x})$ of piece-wise linear models (cf. Sec. 3.1) in color would not seem possible.

However, we note that in combination with the commonly used input *normalisation*, we can convert the first linear transformation in conventional models (e.g., a convolutional layer) into an equivalent transformation that accepts 6-channel inputs. Specifically, for input normalisation, the channel-wise means $\mu_s$ are subtracted from the individual channels, followed by a division by the standard deviations $\sigma_s$, yielding $s' = (s - \mu_s)/\sigma_s$ for $s \in \{r, g, b\}$. Conversely, mean-normalising the 3 additional color channels yields $-s'$. Leveraging this, we use the models' weights learnt for 3-channel inputs, $\mathbf{w}_j = [w_{j,r}, w_{j,g}, w_{j,b}]$ for every feature $j$, to construct an equivalent 6-channel transformation:

$$\mathbf{w}_j' = \left[ \frac{\mathbf{w}_{j,r}}{2}, \frac{\mathbf{w}_{j,g}}{2}, \frac{\mathbf{w}_{j,b}}{2}, -\frac{\mathbf{w}_{j,r}}{2}, -\frac{\mathbf{w}_{j,g}}{2}, -\frac{\mathbf{w}_{j,b}}{2} \right] \, . \tag{2}$$

Note that applying $\mathbf{w}_j'$ to the mean-normalised, 6-channel inputs yields the same results as applying $\mathbf{w}_j$ to the original mean-normalised inputs that the pre-trained models have seen during training.

**Activation Functions.** Owing to the non-linearity inherent to the B-cos transform, explicit activation functions are not necessary in between B-cos layers. However, the authors of [10, 11] showed that the model-inherent explanations are compatible with MaxOut [21]. Note that the very commonly used ReLU non-linearity applied to $\mathbf{v}^T\mathbf{x}$ for any weight vector $\mathbf{v}$, is just a special case of MaxOut:

$$\text{MaxOut}(\mathbf{x}; \mathbf{v}, \mathbf{0}) = \max(\mathbf{v}^T\mathbf{x}, \mathbf{0}^T\mathbf{x}) = \text{ReLU}(\mathbf{v}^T\mathbf{x}) \, . \tag{3}$$

As the pre-trained models' weights have been optimised for the ReLU non-linearity and given its compatibility with the B-cos explanations, we leave them untouched in the B-cosification process.

**Weight normalization.** B-cos transformations employ unit norm weights, see also Eq. (1), which the authors motivated by the fact that the only way any given neuron can achieve its maximal output is by increasing the weight-input alignment, which in turns leads to the improvements of the explanations.

However, conventional models have been trained with unconstrained weights and using unit norm weights would thus lead to unpredictable model behaviour. Interestingly, we note that the weight normalisation in the latest version of the B-cos models can actually not impact the explanation quality, as the authors of [11] re-introduce normalisation layers into the B-cos models. To better understand this, let us consider the compound function of a batch normalisation layer and a B-cos layer:

$$f(\mathbf{x}) = \text{BatchNorm} \circ \text{B-cos}(\mathbf{x}) \tag{4}$$

$$\text{with} \quad \text{BatchNorm}(\mathbf{y}) = \alpha \times \frac{\mathbf{y} - \text{mean}(\mathbf{y})}{\sqrt{\text{var}(\mathbf{y})}} + \beta \, , \tag{5}$$

with $\alpha$ and $\beta$ trainable parameters of the BatchNorm layer. Note that $\sqrt{\text{var}}$ is scaled by any factor $\gamma$ by which the output of a B-cos layer might be scaled, which cancels in the fraction in Eq. (5) and thus makes $f(\mathbf{x})$ *invariant* to scaling the B-cos transformation: i.e. $\text{BatchNorm}(\mathbf{y}) = \text{BatchNorm}(\gamma \times \mathbf{y})$.

Table 2: **Increasing $B$ for B-cosification.**

| ResNet-18 / Metric | Baselines | | Discrete B | | | Linear B | | | Learnt B |
|---|---|---|---|---|---|---|---|---|---|
| | Standard | B-cos | B=1 | B=1.5 | B=2 | 5 epo. | 45 epo. | 90 epo. | |
| Accuracy | 69.6±0.2 | 68.5±0.2 | 70.6±0.1 | 71.6±0.1 | 71.5±0.1 | 71.6±0.2 | 71.1±0.1 | 70.2±0.0 | 71.8±0.1 |
| Localisation | 21.4±0.2 | 87.4±0.5 | 33.9±0.2 | 84.3±0.2 | 87.6±0.2 | 88.1±0.1 | 88.8±0.2 | 88.8±0.2 | 89.4±0.1 |

In particular, the output of $\mathbf{f}(\mathbf{x})$ is thus invariant to weight normalisation, as the output of B-cos $(\mathbf{x})$ scales linearly with the weight norm, cf. Eq. (1).

This is of course only true if every B-cos layer were always followed by a normalisation layer, which is not necessarily the case. Nonetheless, we find that not using normalised weights yields consistently good results across all models. Therefore, we use B-cos transformations without weight normalisation throughout our experiments; for an ablation, see Tab. B2 in the appendix.

**In summary,** we showed that it is possible to adapt the implementation of existing models in a way that allows us to integrate certain aspects of B-cos models without functionally changing the pre-trained models. Notably, we can now visualise color explanations similar to B-cos models (Fig. 1, right, col. 3); unsurprisingly, however, these explanations have poor interpretability due to the absence of the alignment pressure imposed during B-cos training. In the next section, we discuss the necessary functional changes for B-cosification to obtain interpretable explanations.

### 3.2.2 Fine-tuning for Interpretability

The changes introduced in the preceding section have not functionally changed the pre-trained models, but rather allow us to interpret the existing models as a special case of B-cos models. Now we introduce the necessary changes to increase the interpretability of the dynamic matrices $\mathbf{W}(\mathbf{x})$. As these functionally change the models, they need to be fine-tuned to recover their original performance.

In particular, the remaining differences between conventional and B-cos models are (1) the value of $B$, and (2) the use of biases, Tab. 1. We will now discuss how we bridge these differences individually.

**Ablation Setup.** We evaluate various fine-tuning strategies using a ResNet-18 [22] model supervised on ImageNet [16] from Torchvision [54] for B-cosification, and compare with a B-cos ResNet-18 from [11]. We optimize using AdamW [26] with cosine scheduling and train for 90 epochs, and evaluate both classification accuracy as well as interpretability using the GridPG metric [9].

**(1) Increasing $B$.** As shown in [10], using $B>1$ is critical to obtain easily interpretable explanations. To increase B for the pre-trained models, we investigate three strategies: (1) immediately setting $B$ to a higher value and then fine-tuning, (2) linearly interpolating from $B = 1$ to $B = 2$ throughout fine-tuning, and (3) setting $B$ as a learnable parameter. (2) has the advantage of changing the model in small steps, making it more likely that it maintains performance while fine-tuning, but requires using the full number of epochs to reach the target value of $B$. (1) on the other hand is likely to adversely affect the utility of the weights, but offers the opportunity to stop fine-tuning early if performance and interpretability metrics are sufficiently high. (3) offers the most flexibility, but also adds a new set of parameters that need to be optimized. We show the results of this evaluation in Tab. 2. Interestingly, we find that using (1), i.e. setting $B = 2$ and then fine-tuning, yields performance that is on par with learnable B parameters, whilst being significantly simpler to implement. To easily test the generality of the B-cosification scheme, we therefore opt for this approach in Sec. 4.1.

**(2) Decreasing biases.** As discussed in Sec. 3.1, dynamic linear models with bias terms are not *exactly* summarised by the matrix $\mathbf{W}(\mathbf{x})$, cf. [52]. To obtain the same level of *faithfulness* of the explanations as B-cos models (in particular w.r.t. explanation completeness, cf. [52, 53]), we need to remove the biases from the model. To do so, we investigate two approaches: (1) removing all biases first and then fine-tuning, and (2) fine-tuning while decaying biases using weight decay. Similar to the setup with $B$, (2) has the advantage of avoiding drastic changes to the model, but requires potentially fine-tuning for longer. Further, the weight given to the bias decay in the loss constitutes a tradeoff between maintaining classification performance and pushing the biases to be close to zero. We report the results of this evaluation in Tab. 3. Similarly to the experiments for $B$, we find that immediately setting the biases to zero constitutes a simple yet performant approach to achieve both good localisation and accuracy. To assess the generality of the B-cosification scheme across a wide range of models, we thus choose this the simpler approach of setting biases to zero in Sec. 4.1.

Table 3: **Decreasing biases for B-cosification.**

| ResNet-18 / Metric | Baselines | | Fixed bias | | | Bias decay | | |
|---|---|---|---|---|---|---|---|---|
| | Standard | B-cos | With bias | No bias | decay=0.2 | decay=0.5 | decay=0.9 |
| Accuracy | $69.6_{\pm0.2}$ | $68.5_{\pm0.2}$ | $71.2_{\pm0.1}$ | $71.5_{\pm0.1}$ | $71.2_{\pm0.2}$ | $71.4_{\pm0.3}$ | $71.6_{\pm0.2}$ |
| Localisation | $21.4_{\pm0.2}$ | $87.4_{\pm0.5}$ | $47.2_{\pm0.5}$ | $87.6_{\pm0.2}$ | $81.4_{\pm0.2}$ | $90.2_{\pm0.2}$ | $91.2_{\pm0.1}$ |

In short, we find that a very simple approach, i.e., setting the bias and the $B$ values to the target values immediately, constitutes a simple and easy-to-use, but nonetheless performant strategy to B-cosify models. In the following sections, we test whether these findings generalise well to other models.

## 4 B-cosification Results

In the following, we evaluate the effectiveness of the B-cosification strategy we developed in Sec. 3. In Sec. 4.1, we first apply B-cosification to supervised models across various architectures, and evaluate for classification performance and interpretability. In Sec. 4.2, we B-cosify CLIP [39], a large foundation vision-language model, and show that despite fine-tuning at a fraction of the training cost, the B-cosified CLIP shows strong zero shot generalization whilst being highly interpretable.

### 4.1 Supervised Classification Models

Table 4: **Classification Accuracy.** We report the top-1 classification accuracy on the ImageNet validation set of the pre-trained models (**pretrained**) and the B-cosified models (**B-cosified**) after fine-tuning them. Additionally, we report the accuracy of the corresponding B-cos models trained from scratch (**B-cos**) as well as the difference to them ($\Delta_{\text{acc}}$), and how much faster and at which epoch ($t$) the same accuracy as in [11] was achieved (**speedup**). Results for B-cosified models are averaged over three runs; full results including standard deviation in appendix.

| Model | Top-1 Accuracy (%) | | | | Efficiency Gains | |
|---|---|---|---|---|---|---|
| | pretrained | B-cos [11] | B-cosified | $\Delta_{\text{acc}}$ | $t$ | speedup |
| ResNet-18 | 69.8 | 68.7 | 71.5 | +2.8 | 29 | ×3.1 |
| ResNet-50-v1 | 76.1 | 75.9 | 76.5 | +0.6 | 46 | ×2.0 |
| ResNet-50-v2 | 80.9 | 75.9 | 77.3 | +1.4 | 10 | ×9.0 |
| DenseNet-121 | 74.4 | 73.6 | 76.3 | +2.7 | 18 | ×5.0 |
| ViT-Ti | 70.3 | 60.0 | 69.3 | +9.3 | 10 | ×9.0 |
| ViT-S | 74.4 | 69.2 | 75.2 | +6.0 | 10 | ×9.0 |
| ViT-B | 75.3 | 74.4 | 75.3 | +0.9 | 57 | ×1.6 |
| ViT-L | 75.8 | 75.1 | 75.5 | +0.4 | 66 | ×1.4 |
| $\text{ViT}_c$-Ti | 72.6 | 67.3 | 72.3 | +5.0 | 10 | ×9.0 |
| $\text{ViT}_c$-S | 75.7 | 74.5 | 76.0 | +1.5 | 32 | ×2.8 |
| $\text{ViT}_c$-B | 76.8 | 77.1 | 76.7 | -0.4 | - | - |
| $\text{ViT}_c$-L | 77.9 | 77.8 | 77.1 | -0.7 | - | - |

**Setup.** We B-cosify models from Torchvision [54] supervised on ImageNet [16]. We use a diverse set architectures, including both CNNs (ResNet-18 [22], ResNet-50 [22], and DenseNet-121 [24]), and ViTs [18, 6, 57] with ($\text{ViT}_c$-Ti, $\text{ViT}_c$-S, $\text{ViT}_c$-B, $\text{ViT}_c$-L) and without (ViT-Ti, ViT-S, ViT-B, ViT-L) convolutional stems. For ResNet-50, we use both the weights originally released by Torchvision and the updated V2 weights, which constitute models trained for longer and with more augmentations [55]. As in Sec. 3.2, we evaluate both for classification accuracy and for interpretability using the GridPG [9] metric. We compare both accuracy and interpretability of the B-cosified models with B-cos models trained from scratch from [11]. For interpretability, we also compare with several post-hoc attribution methods as baselines, namely Guided Backprop [51], Gradient [49], DeepLIFT [48], IxG [48], IntGrad [53], and GradCAM [46]. For full details, see Appendix C.1.

**Classification performance.** Tab. 4 reports the classification accuracy of the B-cosified models across architectures, and compares them with their conventional counterparts from Torchvision and B-cos models trained from scratch. We find that across architectures (col. 1), B-cosified models perform competitively with conventional DNNs (cols. 2-4) and interestingly, in contrast to the findings reported by [11], often outperform them, i.e. for five out of twelve architectures. Notably, we find (col. 5) that our B-cosified models significantly outperform B-cos models trained from

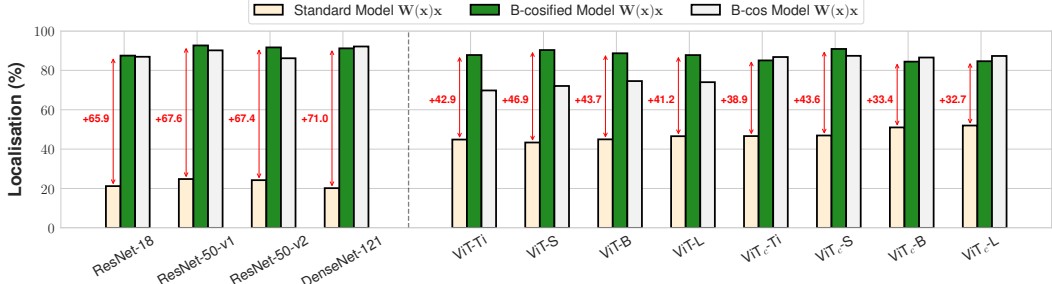

Fig. 3: **Localisation Performance of $\mathbf{W}(\mathbf{x})\mathbf{x}$.** We compute the contribution maps according to the dynamic linear summaries $\mathbf{W}(\mathbf{x})$ of the pre-trained models ('Standard'), their B-cosified versions, and the original pre-trained B-cos models and evaluate their localisation performance on the Grid Pointing Game as in [11]. We find localisation to significantly improve for B-cosified models, achieving results on par with the models of [11].

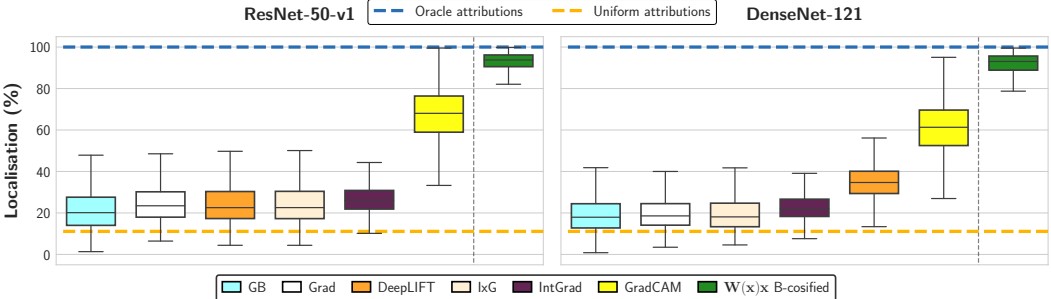

Fig. 4: **Comparison to Post-hoc Methods.** For two of the models in Fig. 3 (ResNet-50-v1, DenseNet-121) we compare the localisation performance of the dynamic matrices $\mathbf{W}(\mathbf{x})\mathbf{x}$ to post-hoc explanations for the pre-trained models. Similar to the original B-cos models [11], the model-inherent explanations perform favourably.

scratch across all but the two largest ViT$_c$ architectures. Further, B-cosified models often achieve the same performance as their corresponding B-cos models at a fraction of the training cost (col. 6). Specifically, we find that averaged across architectures, B-cosified models outperform B-cos models trained from scratch by 2.5 pp, with an average training speedup (to match performance) of 5.2x. These results strongly advocate for B-cosification as a superior alternative to training from scratch for obtaining performant inherently interpretable models at a low compute cost.

**Interpretability.** To evaluate the interpretability of our B-cosified models, we report the GridPG localization scores in Fig. 3, and compare with conventional and B-cos models; following [11], we report the results of 3x3 image grids for convolutional models, and of 2x2 grids for the ViTs. For a fair comparison, for all models, we evaluate the localization of the dynamic linear summary of the model[2] $\mathbf{W}(\mathbf{x})\mathbf{x}$ (see Sec. 3.1). We find that across architectures, B-cosified models significantly outperform conventional DNNs in terms of localization (32.7pp-71.0pp) and perform on par with B-cos models. Since post-hoc attribution methods (e.g. [46, 48, 53]) are often used to interpret conventional DNNs, similar to [10], in Fig. 4, we compare the localization of the model inherent explanations from two of our B-cosified models with post-hoc explanations applied to the corresponding conventional models. Similar to the results reported by [10], we find our B-cosified models to strongly outperform all post-hoc methods, including GradCAM [46], with a near perfect localization score, showing that B-cosification is effective in yielding highly interpretable yet model-faithful explanations.

**Impact of pre-trained weights.** Since our aim is to *fine-tune* for interpretability, we investigate how crucial the quality of the weights of the conventional model are for effective B-cosification. Specifically, we expect weights from stronger models to be a better starting point for B-cosification. We evalaute this by performing B-cosification both with v1 and v2 variants of ResNet-50 [22] from Torchvision [54], where the latter is trained for longer and with stronger augmentations [55]. From Tab. 4, we find that using a strong initialization is highly useful for effective B-cosification,

---

[2]Note that ViTs differ from the CNNs discussed in Tab. 1 via the attention mechanism and the GELU activation with $\mathrm{GELU}(x) = x \times (0.5 + 0.5 \times \mathrm{erf}(x/\sqrt{x})$. As attention is also dynamic linear, cf. [11], it can seamlessly be integrated into the model summary $\mathbf{W}(\mathbf{x})$. Similarly, we interpret the second factor in GELU as a dynamic weight $w(x)$, thus allowing us to integrate it in a similar fashion.

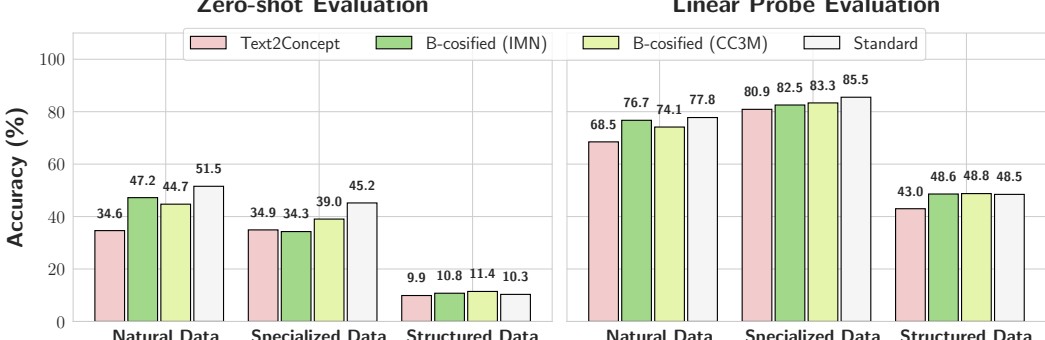

Fig. 5: **Classification performance on the CLIP Benchmark** [15] of various CLIP models for the zero-shot setting (*left*) and linear probing (*right*). Specifically, we compare two B-cosified CLIPs—trained on ImageNet (IMN) and CC3M respectively—to the Text2Concept approach by [30] and the original pre-trained CLIP model. We find B-cosified versions of CLIP to consistently outperform Text2Concept on natural and specialised data.

specifically, fine-tuning starting from v2 weights outperforms B-cos models by 0.8pp as compared to v1, and achieves equal accuracy with a 9x speedup as compared to 2x with v1; similar results are observed for initialising the weights with models pretrained via the self-supervised DINO paradigm [12] (final accuracy: 77.0, speedup: 3.2x), for further discussion see Tab. B3 in the appendix.

## 4.2  B-cosifying CLIP — Towards Inherently Interpretable Foundation Models

In this section, we evaluate our B-cosification paradigm on CLIP [39], a powerful pre-trained vision-language model, and evaluate its interpretability and zero shot performance.

**Setup.** We B-cosify a CLIP [39] with a ResNet-50 [22] backbone using the procedure described in Sec. 3.2. We use the recently proposed SigLIP loss [59] between the image embeddings of the pre-trained CLIP and the B-cosified CLIP's and train the models on either the ImageNet [16] or the CC3M datasets [47]. For evaluation, we rely on the CLIP Benchmark [15] and report zeroshot and linear probing results for accuracy. To assess the models' interpretability, we explain the similarity between the image embeddings and the text embedding of the pre-trained CLIP model via the dynamic linear summaries, see Sec. 3.1 or GradCAM, and report the EPG scores [56, 41] on the VOC dataset [19]. For full details, see Appendix C.2.

**Evaluating Model Performance.** In Fig. 5, we report the zeroshot and linear probing accuracies of the two B-cosified CLIP models (trained on ImageNet or CC3M) and compare it to the original CLIP (Standard) and the recently proposed Text2Concept technique [30]; for the latter, we train a linear layer on top of a frozen, pre-trained B-cos ResNet-50 from [11] to mimic the embeddings of CLIP [30]. We find that the B-cosified models significantly outperform the Text2Concept approach and achieve accuracies that are more similar to the original CLIP's zeroshot and linear probing accuracies.

**Evaluating Model Interpretability.** We evaluate the B-cosified CLIP's ability to localise classes in the VOC dataset in two ways. On the one hand, we directly explain the similarity of the models' embedding to the text embedding of a given prompt such as "A photo of a cow.". On the other hand, we note that the final attention pooling layer in the CLIP model only computes a weighted sum of the last layer's value vectors. Therefore, we additionally evaluate whether we can also explain the similarity between the text embeddings and these value vectors to improve the localisation ability.

In this context, we notice that explaining the average similarity to the text embedding yields highly distributed attribution maps, see Fig. 6b, col. 2. On the other hand, explaining only the most similar embedding localises very well, see Fig. 6b, col. 5. To better understand this phenomenon, we additionally interpolate between these two approaches and compute *weighted means* $\sum_i w_i \mathbf{v}_i$ of those value vectors $\mathbf{v}_i$, in which the weights are determined by the cosine similarity between the value vectors $\mathbf{v}_i$ and the text embedding $\mathbf{t}$, i.e. with weights $w_i = \cos^p(\mathbf{t}, \mathbf{v}_i)$ for various $p$.

We find that this not only significantly improves the explanations qualitatively, see Figs. 2 and 6b, but also quantitatively: in Fig. 6a we report results for explaining the final image embedding (**B-cosified CLIP**), the dynamic linear summary for the CLIP ResNet-50 (**CLIP $\mathbf{W}(\mathbf{x})\mathbf{x}$**), its GradCAM explanations (**CLIP GradCAM**), and the weighted mean of the value vectors, which we call

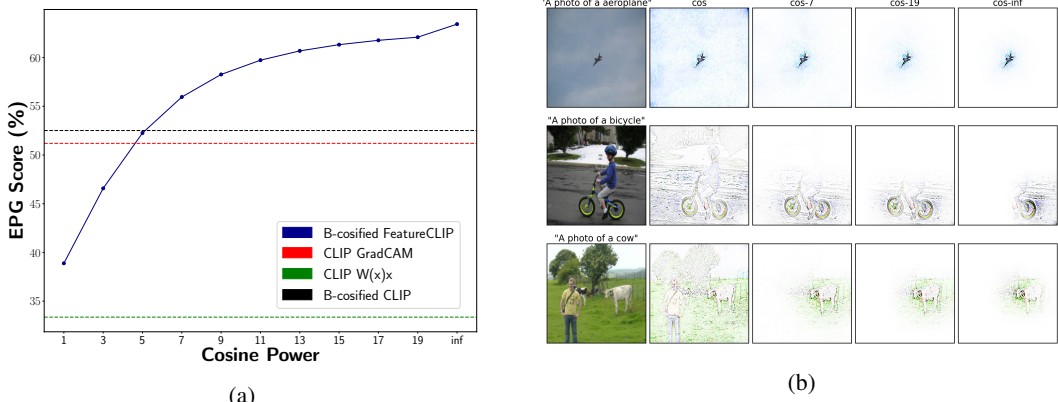

(a)                                          (b)

Fig. 6: **CLIP Localisation.** In (a), we compare GradCAM and dynamic linear explanations for the pre-trained CLIP model to the inherent explanations of the B-cosified CLIP, as well as to our proposed B-cos FeatureCLIP approach. We find that good localisation with highly detailed explanations (b) is possible via the B-cosified CLIP models, especially for high cosine powers, despite only explaining the similarity to text prompts.

**B-cosified FeatureCLIP**. While B-cos CLIP already yields a noticeable improvement over the pre-trained CLIP explained via GradCAM and significantly improves the localisation ability of the linear summary $\mathbf{W(x)x}$, the weighted means yield even stronger performance for sufficiently high $p$.

## 5  Discussion

The B-cosification approach presented in this work addresses a common issue with developing inherently interpretable models: achieving model interpretability without compromising on performance or incurring high training costs. By leveraging pre-trained models, B-cosification opens a new path towards developing interpretable yet performant models, which can be of particular interest in the context of foundation models such as CLIP [39], which might otherwise be prohibitively expensive to train on a limited budget. Our results suggest that B-cosification not only maintains but, in several cases, even enhances model accuracy, whilst yielding significant improvements on interpretability metrics, providing a viable and resource-efficient alternative to training B-cos models from scratch.

Specifically, we find B-cosified models to much faster reach the same levels of interpretability and accuracy than their counterparts trained from scratch, with training speedups of up to 9x in some models. The approach appears to be general, being applicable for both CNNs and ViT models. We hope that this increase in efficiency will make interpretable models much more accessible in settings with constrained computational resources and could thus facilitate their adoption. In particular, when applying our proposed B-cosification scheme to a foundation model—CLIP—we find that the B-cosified CLIP model is able to maintain competitive zero-shot performance while at the same time providing interpretable and model-faithful explanations.

Despite these advancements, certain aspects remain open for further exploration. Specifically, while some models quickly recover original performance after B-cosification, others exhibit slower convergence rates, suggesting potential for optimisations in the fine-tuning process. Additionally, for the larger B-cosified ViT$_c$ models, while yielding results that are on par with those trained from scratch, the B-cosification process did not succeed in achieving speed-ups, indicating that the interplay between model architecture and the proposed B-cosification might require further exploration.

In summary, our results establish B-cosification as an effective method for enhancing interpretability in pre-trained models with low computational cost. The method consistently enables high interpretability without compromising performance, even achieving substantial training speedups in many cases.

## Acknowledgements

Funded in part by the Deutsche Forschungsgemeinschaft (DFG, German Research Foundation) – GRK 2853/1 "Neuroexplicit Models of Language, Vision, and Action" - project number 471607914.

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

# B-cosification: Transforming Deep Neural Networks to be Inherently Interpretable

# Appendix

## Table of Contents

In this supplement to our work on B-cosification of black box models, we provide:

# A   Additional Qualitative Results

In Fig. A1, we provide additional qualitative examples to illustrate the interpretability gains achieved by B-cosifying a CLIP model. Specifically, we show explanations generated by the original CLIP model using GradCAM [46] (row 2) for a diverse set of input images (row 1), for which explanations are generally coarse and lack clear localization. In contrast, the third row displays explanations produced by B-cosified CLIP, which yields finer-grained, more visually interpretable explanations.

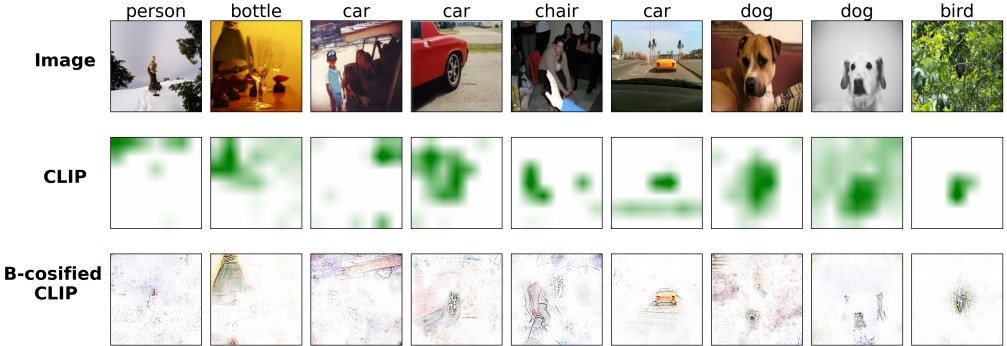

Fig. A1: Additional, randomly sampled examples for comparing GradCAM explanations of the original CLIP model to the inherent explanations of the B-cosified CLIP. The top row shows input images from various classes. The middle row provides explanations generated by the original CLIP model, which tend to be coarse and lack precise localization. The bottom row shows explanations from B-cosified CLIP, which produce more focused, detailed visualizations, highlighting class-relevant features with greater clarity.

In Fig. A2, we show further comparisons on specific object classes with using different cosine powers $p$ (cos, cos-7, cos-19, and cos-inf) to qualitatively demonstrate the effect of increasing the exponent $p$ in gathering the value vectors, see also Sec. 4.2. Higher cosine thresholds result in increasingly focused and interpretable representations, capturing fine details that are often absent in the original CLIP explanations.

In Fig. A3, we show additional qualitative examples for prompting the B-cosified CLIP model with different prompts for the same image, thus highlighting the class-specificity of the explanations as well as the potential that inherently interpretable CLIP models might yield. Specifically, B-cosified CLIP models allow to explain the similarity of a given image with a free-form textual prompt, which shows that the zero-shot performance of CLIP with respect to classification also transfers well to the corresponding explanations.

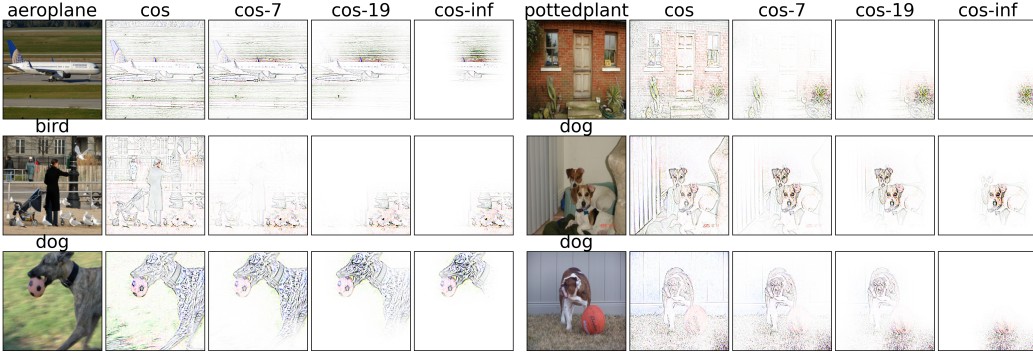

Fig. A2: Additional randomly chosen examples highlighting the effect of increasing cosine power $p$ on the specificity of the explanations in the B-cosified CLIP model. Each row corresponds to a specific object class, with explanations generated at different cosine power levels: cos, cos-7, cos-19, and cos-inf. Higher cosine power values result in increasingly precise and interpretable representations, capturing finer details and producing sharper focus on class-relevant features; further examples in Fig. 6b in the main paper.

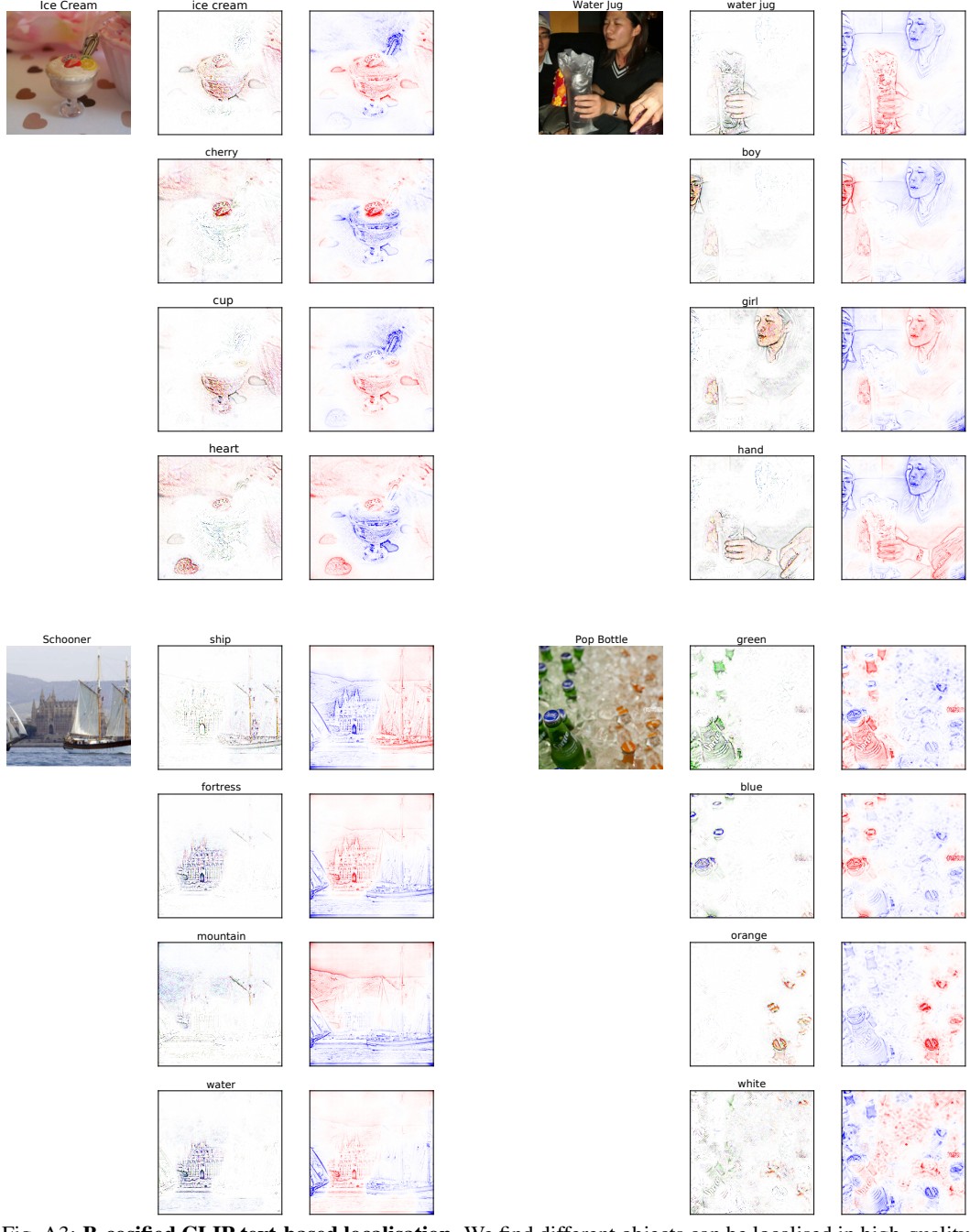

Fig. A3: **B-cosified CLIP text-based localisation**. We find different objects can be localised in high-quality color using B-cosified CLIP. For example, we compute B-cos explanations using text-prompts: a picture of {'x'} encoded using standard CLIP [39] text-encoder for an image taken from ImageNet [45]. The first column shows the original image with the class caption, the second column shows the colored B-cos explanations with the text caption used at the top, and the third column shows the raw attributions with red denoting the positive contribution towards the text input and blue denoting the negative contribution.

# B    Additional Quantitative Results

In this section, we provide a series of additional quantitative results on the performance and interpretability of B-cosified models. These tables cover various ablation studies, comparisons with standard and B-cos models, and performance across different configurations. Specifically, in Tab. B1, we extend our evaluation to compare models that are trained for the same effective number of epochs[3]. Further, in Tab. B2, we evaluate the impact of using normalized weights in the B-cos layers. In Tab. B3, we show additional results for B-cosified ResNet-50 models that are initialised from different pre-trained checkpoints. Specifically, we find that apart from initialising the ResNet-50 from CLIP weights, we observe consistent improvements through B-cosification, with stronger pre-training paradigms that are based on the ImageNet dataset (V2, DINO [12]) leading to larger improvements. Finally, we report full ablation results on the impact of the individual changes that we perform on the pre-trained models Tab. B4, B-cosifying models with different strategies for setting the parameter B in the B-cos transformation Tab. B5, decaying the bias term Tab. B6, as well as full zero-shot and linear probing results for the CLIP benchmark, see Tabs. B7 and B8.

Table B1: **Classification accuracy comparison with further trained standard and B-cos models.** Extending Tab. 4, we report the comparison of top-1 classification accuracy on the ImageNet validation set between the **B-cosified** models and comparison with standard (block 3) and B-cos (block 4) pre-trained models fine-tuned using the same process as B-cosification. All models are thus effectively trained for 180 epochs. Results for B-cosified models are averaged over three runs.

**Training for 180 Epochs**

| Model | B-cosified acc | Standard models acc | $\Delta_{\text{acc}}^1$ | speedup | B-cos models acc | $\Delta_{\text{acc}}^2$ | speedup |
|---|---|---|---|---|---|---|---|
| ResNet-18 | $71.5_{\pm 0.1}$ | 71.0 | +0.3 | $\times 1.4_{\pm 0.1}$ | 68.9 | +2.4 | $\times 2.8_{\pm 0.0}$ |
| ResNet-50-v1 | $76.5_{\pm 0.1}$ | 76.8 | -0.3 | - | 75.6 | +0.9 | $\times 2.4_{\pm 0.1}$ |
| ResNet-50-v2 | $77.3_{\pm 0.1}$ | 77.7 | -0.4 | - | 75.6 | +1.7 | $\times 10.4_{\pm 0.7}$ |
| DenseNet-121 | $76.3_{\pm 0.2}$ | 76.1 | +0.2 | $\times 1.4_{\pm 0.1}$ | 73.8 | +2.5 | $\times 5.2_{\pm 0.4}$ |
| ViT-Ti | $69.3_{\pm 0.1}$ | 72.1 | -2.8 | - | 63.7 | +5.6 | $\times 3.4_{\pm 0.1}$ |
| ViT-S | $75.2_{\pm 0.1}$ | 76.4 | -1.2 | - | 72.5 | +2.7 | $\times 2.7_{\pm 0.0}$ |
| ViT-B | $75.3_{\pm 0.1}$ | 75.8 | -0.5 | - | 75.3 | - | - |
| ViT-L | $75.5_{\pm 0.1}$ | 75.7 | -0.2 | - | 75.6 | -0.1 | - |
| ViT$_c$-Ti | $72.3_{\pm 0.1}$ | 74.8 | -2.5 | - | 70.1 | +2.2 | $\times 1.8_{\pm 0.0}$ |
| ViT$_c$-S | $76.0_{\pm 0.1}$ | 76.9 | -0.9 | - | 75.9 | +0.1 | $\times 1.3_{\pm 0.1}$ |
| ViT$_c$-B | $76.9_{\pm 0.2}$ | 77.2 | -0.3 | - | 77.2 | -0.3 | - |
| ViT$_c$-L | $77.1_{\pm 0.0}$ | 77.6 | -0.5 | - | 77.7 | -0.6 | - |

---

[3]B-cosified models have effectively been trained for 180 epochs, i.e., 90 epochs of pre-training and 90 epochs of fine-tuning. To fairly compare the impact of the additional training, we show how this would impact both the pre-trained models themselves, as well as the B-cos models from [11]. Note that however the primary goal is to leverage pre-trained weights to more efficiently train B-cos models, and as shown in Tab. 4, B-cosification helps obtaining similarly accurate and interpretable models at a much lower training cost.

Table B2: **Ablation normed weights for the ViT models.** Extending Tab. 4 for a subset of models, we report the top-1 classification accuracy on the ImageNet validation set of the pre-trained models (**pretrained**) and the B-cosified models (**B-cosified**) after fine-tuning them along with the difference between them ($\Delta^1_{\text{acc}}$). Additionally, we report the accuracy of the corresponding B-cos models trained from scratch (**B-cos**) as well as the difference to them ($\Delta^2_{\text{acc}}$), and how much faster and at which epoch ($t$) the same accuracy as in [11] was achieved (**speedup**). Results for B-cosified models are averaged over three runs.

| Model | Top-1 Accuracy (%) | | | Gains over B-cos [11] | | | |
| | pretrained | B-cosified | $\Delta^1_{\text{acc}}$ | B-cos | $\Delta^2_{\text{acc}}$ | $t$ | speedup |
|---|---|---|---|---|---|---|---|
| **Un-norm wt.** | | | | | | | |
| ViT-Ti | 70.3 | 69.3±0.1 | -1.1 | 60.0 | +9.3 | 10 | ×9.0 |
| ViT-S | 74.4 | 75.2±0.1 | +0.8 | 69.2 | +6.0 | 10 | ×9.0 |
| ViT-B | 75.3 | 75.3±0.1 | -0.1 | 74.4 | +0.9 | 57 | ×1.6 |
| ViT$_c$-Ti | 72.6 | 72.3±0.1 | -0.3 | 67.3 | +5.0 | 10 | ×9.0 |
| ViT$_c$-S | 75.7 | 76.0±0.1 | +0.3 | 74.5 | +1.5 | 32 | ×2.8 |
| ViT$_c$-B | 76.8 | 76.7±0.2 | +0.1 | 77.1 | -0.4 | - | - |
| **Norm wt.** | | | | | | | |
| ViT-Ti | 70.3 | 68.5±0.2 | -1.8 | 60.0 | +8.5 | 26 | ×3.5 |
| ViT-S | 74.4 | 76.0±0.2 | +1.6 | 69.2 | +6.8 | 24 | ×3.8 |
| ViT-B | 75.3 | 76.9±0.1 | +1.6 | 74.4 | +2.5 | 30 | ×3.0 |
| ViT$_c$-Ti | 72.6 | 73.2±0.2 | +0.6 | 67.3 | +5.9 | 28 | ×3.2 |
| ViT$_c$-S | 75.7 | 78.0±0.1 | +2.3 | 74.5 | +3.5 | 27 | ×3.3 |
| ViT$_c$-B | 76.8 | 78.3±0.1 | +1.5 | 77.1 | +1.2 | 32 | ×3.8 |

Table B3: **Impact of Pre-trained Weights.** We report the top-1 classification accuracy on the ImageNet validation set of the **B-cosified ResNet-50** model with different weight initialisations (**acc**), the difference to the results reported in [11] when training the pre-trained B-cos ResNet-50 model ($\Delta_{\text{acc}}$) under the same B-cosification recipe, as well as how much faster the same accuracy was achieved (**speedup**). Additionally, we report the GridPG localisation scores (**loc**) similar to those reported in [11] and the difference to the B-cos ResNet-50 model's localisation trained ($\Delta_{\text{loc}}$) under the same B-cosification training recipe. The pre-trained accuracy (in %) for: B-cos = 75.9, CLIP = 73.3, and DINO = 75.3. Random Init is the baseline with randomly initialized B-cosified model training. Results are for a random initialized single run.

| Weights | Top-1 Accuracy (%) | | | Localisation [11] | |
| | acc | $\Delta_{\text{acc}}$ | speedup | loc | $\Delta_{\text{loc}}$ |
|---|---|---|---|---|---|
| Random Init | 72.7 | -2.9 | - | 88.8 | -1.6 |
| V1 | 76.6 | +1.0 | ×2.4 | **92.4** | +2.0 |
| V2 | **77.3** | +1.7 | ×11.3 | 91.7 | +1.3 |
| CLIP | 75.3 | -0.3 | - | 91.2 | +0.8 |
| DINO | 77.0 | +1.4 | ×3.2 | 90.9 | +0.5 |

Table B4: **Experimental ablations results for the B-cosification design choices.** The table shows the ablation results for the **B-cosified ResNet-18** model (with $B = 2$ and no Bias) shown in row 1. Further in rows 2-6, the ablated components are shown in col. 1, the accuracy of the ablated model (**acc**) in col. 2, and the change in accuracy ($\Delta_{acc}$) compared to the non-ablated model (row 1) is shown in col. 3. Further, col. 4 shows the GridPG localisation scores (**loc**) and the difference to the non-ablated B-cosified model ($\Delta_{loc}$). Similar to components used in standard models, we replace the BatchNorm Uncentered with BatchNorm Centered, change the order of the Global Average Pool where features are first averaged and then passed to the last linear layer, and replace average pooling with max pooling at the stem. Further, we also replaced ReLU activation with Identity; and removed the Logit Bias layer. Results are for a random initialized single run.

| | Top-1 Accuracy (%) | | Localisation | |
|---|---|---|---|---|
| **Ablated Component** | acc | $\Delta_{acc}$ | loc | $\Delta_{loc}$ |
| B-cosified | 71.5 | - | 86.8 | - |
| BatchNorm Centered | 71.0 | -0.5 | 28.8 | -58.0 |
| Global Average Pool Order | 71.5 | 0.0 | 63.8 | -23.0 |
| Identity Activation | 63.4 | -8.1 | 86.0 | -1.4 |
| Logit Bias removal | 70.8 | -0.7 | 92.6 | +5.2 |
| Max Pool (Stem) | 71.4 | -0.1 | 87.7 | +0.3 |

Table B5: **Increasing $B$ for B-cosification Extended.** Extending Tab. 2 for different convolutional models, we compare various strategies to increase the $B$ parameter. We evaluate these strategies using accuracy and localization scores to measure interpretability performance. Columns 2-3 represent the baseline models: Standard ResNet-18 [54] and B-cos ResNet-18 [11]. Columns 4-8 set the $B$ value directly $\in \{1, 1.25, 1.5, 1.75, 2\}$. Columns 9-13 apply $B = 2$ over 'n' epochs with $n \in \{5, 10, 20, 45, 90\}$. Column 14 shows results for increasing $B$ as a learned parameter. Further, row blocks denote different models, each denoting accuracy, followed by localisation scores. Results are for a random initialized single run.

| Model / Metric | Baselines | | Discrete B | | | | | Linear B | | | | | Learnt B |
|---|---|---|---|---|---|---|---|---|---|---|---|---|---|
| | Standard | B-cos | B=1 | B=1.25 | B=1.5 | B=1.75 | B=2 | 5 epo. | 10 epo. | 20 epo. | 45 epo. | 90 epo. | |
| **ResNet-18** | | | | | | | | | | | | | |
| Accuracy | 69.8 | 68.7 | 70.7 | 71.5 | 71.6 | 71.6 | 71.5 | 71.6 | 71.4 | 71.3 | 71.1 | 70.2 | 71.7 |
| Localisation | 21.2 | 88.0 | 33.8 | 68.1 | 84.2 | 86.6 | 87.4 | 88.0 | 87.9 | 88.6 | 88.8 | 89.0 | 89.3 |
| **ResNet-50 V1** | | | | | | | | | | | | | |
| Accuracy | 76.1 | 75.9 | 76.49 | 76.8 | 76.8 | 76.4 | 76.56 | 76.5 | 76.4 | 76.5 | 76.3 | 76.2 | 76.5 |
| Localisation | 24.8 | 90.4 | 45.8 | 86.4 | 91.4 | 92.0 | 92.4 | 92.9 | 92.8 | 92.9 | 92.9 | 92.7 | 93.6 |
| **ResNet-50 V2** | | | | | | | | | | | | | |
| Accuracy | 80.9 | 75.9 | 77.7 | 77.5 | 77.6 | 77.5 | 77.3 | 77.3 | 77.5 | 77.4 | 77.1 | 77.2 | 77.6 |
| Localisation | 24.2 | 90.4 | 46.1 | 86.9 | 91.2 | 91.9 | 91.7 | 91.8 | 92.1 | 92.3 | 92.7 | 92.8 | 92.8 |
| **DenseNet-121** | | | | | | | | | | | | | |
| Accuracy | 74.4 | 73.6 | 75.83 | 76.3 | 76.4 | 76.6 | 76.4 | 76.4 | 76.4 | 76.4 | 76.3 | 75.8 | 76.5 |
| Localisation | 20.2 | 92.3 | 30.4 | 77.0 | 87.0 | 89.9 | 91.2 | 91.4 | 91.7 | 91.8 | 92.2 | 92.2 | 93.6 |

Table B6: **Decreasing biases for B-cosification extended.** Extending Tab. 3 for different convolutional models, we compare various strategies to decrease the bias parameter. We evaluate these strategies using accuracy and localization scores to measure interpretability performance. Columns 2-3 represent the baseline models: Standard ResNet-18 [54] and B-cos ResNet-18 [11]. Columns 4-5 show the setting with bias (col. 4) and without bias (col. 5). Columns 6-8 show the bias decay setup using the weight decay with different values $\lambda \in \{0.2, 0.5, 0.9\}$. Further, row blocks denote different models, each denoting accuracy, followed by localisation scores. Results are for a random initialized single run.

| Model Metric | Baselines | | Fixed bias | | Bias decay | | |
|---|---|---|---|---|---|---|---|
| | Standard | B-cos | With bias | No bias | decay=0.2 | decay=0.5 | decay=0.9 |
| **ResNet-18** | | | | | | | |
| Accuracy | 69.8 | 68.7 | 71.3 | 71.5 | 71.4 | 71.8 | 71.9 |
| Localisation | 21.2 | 88.0 | 46.8 | 87.4 | 81.6 | 90.3 | 91.3 |
| **ResNet-50 V1** | | | | | | | |
| Accuracy | 76.1 | 75.9 | 76.6 | 76.6 | 76.6 | 76.8 | 76.7 |
| Localisation | 24.8 | 90.4 | 92.6 | 92.7 | 93.4 | 93.1 | 92.8 |
| **ResNet-50 V2** | | | | | | | |
| Accuracy | 80.9 | 75.9 | 77.2 | 77.3 | 77.5 | 77.4 | 77.5 |
| Localisation | 24.2 | 90.4 | 88.5 | 91.7 | 93.5 | 93.1 | 93.0 |
| **DenseNet-121** | | | | | | | |
| Accuracy | 74.4 | 73.6 | 76.3 | 76.4 | 76.8 | 76.9 | 76.9 |
| Localisation | 20.2 | 92.3 | 86.9 | 91.2 | 90.0 | 90.3 | 90.7 |

Table B7: **Zero-shot performance of various CLIP-based models** over 38 datasets using CLIP Benchmark [15]. Scores within the 99.5% Clopper-Pearson confidence interval of each dataset's top score are shown in bold. Baselines contain results for the Standard CLIP [39] and Text2Concept (T2C) [30] models; ImageNet and CC3M column sections contain the B-cosified RN-50 CLIP models trained with cosine and cyclic learning schedulers trained with ImageNet [16] and CC3M [47] datasets, respectively. The cyclic learning training inspired from [25]. Dataset type is taken from [15].

| Model / Dataset | Baselines | | ImageNet [16] | | CC3M [47] | |
|---|---|---|---|---|---|---|
| | Standard [39] | T2C [30] | Cosine | Cyclic | Cosine | Cyclic |
| **Natural** | | | | | | |
| cars | **0.54** | 0.02 | 0.37 | 0.38 | 0.35 | 0.36 |
| country211 | **0.15** | 0.03 | 0.12 | 0.12 | 0.12 | 0.13 |
| fer2013 | **0.35** | 0.18 | 0.21 | 0.19 | 0.20 | 0.23 |
| fgvc_aircraft | **0.17** | 0.02 | 0.11 | 0.12 | 0.10 | 0.11 |
| gtsrb | **0.35** | 0.05 | 0.21 | 0.20 | 0.32 | 0.31 |
| imagenet-a | **0.23** | 0.04 | 0.16 | 0.16 | 0.14 | 0.13 |
| imagenet-o | 0.57 | **0.68** | 0.65 | 0.64 | 0.57 | 0.57 |
| imagenet-r | **0.61** | 0.28 | 0.52 | 0.52 | 0.53 | 0.53 |
| imagenet1k | **0.60** | 0.52 | 0.59 | 0.59 | 0.52 | 0.52 |
| imagenet_sketch | **0.35** | 0.15 | 0.28 | 0.28 | 0.29 | 0.30 |
| imagenetv2 | **0.53** | 0.42 | 0.48 | 0.48 | 0.44 | 0.44 |
| objectnet | **0.41** | 0.23 | 0.33 | 0.32 | 0.32 | 0.31 |
| stl10 | **0.94** | 0.91 | **0.94** | **0.94** | 0.93 | 0.93 |
| sun397 | 0.60 | 0.29 | **0.62** | 0.60 | 0.55 | 0.54 |
| voc2007 | **0.65** | **0.65** | 0.63 | 0.62 | 0.62 | 0.61 |
| vtab/caltech101 | **0.77** | 0.74 | 0.74 | 0.74 | 0.72 | 0.72 |
| vtab/cifar10 | 0.71 | 0.35 | 0.71 | 0.71 | 0.71 | **0.72** |
| vtab/cifar100 | 0.40 | 0.09 | 0.40 | **0.41** | 0.37 | 0.36 |
| vtab/dtd | 0.41 | 0.29 | 0.41 | **0.42** | 0.37 | 0.38 |
| vtab/flowers | **0.66** | 0.07 | 0.58 | 0.58 | 0.56 | 0.58 |
| vtab/pets | **0.86** | 0.69 | 0.83 | 0.85 | 0.80 | 0.81 |
| vtab/svhn | **0.30** | 0.08 | 0.11 | 0.15 | 0.13 | 0.14 |
| **Specialized** | | | | | | |
| imagenet_sketch | **0.35** | 0.15 | 0.28 | 0.28 | 0.29 | 0.30 |
| mnist | **0.58** | 0.17 | 0.38 | 0.37 | 0.38 | 0.37 |
| renderedsst2 | **0.56** | 0.50 | 0.50 | 0.50 | 0.50 | 0.50 |
| vtab/diabetic_retinopathy | 0.17 | **0.69** | 0.38 | 0.43 | 0.29 | 0.35 |
| vtab/eurosat | 0.41 | 0.27 | 0.36 | 0.33 | 0.41 | **0.42** |
| vtab/pcam | 0.64 | 0.50 | 0.52 | **0.69** | 0.52 | 0.50 |
| vtab/resisc45 | **0.45** | 0.15 | 0.28 | 0.29 | 0.35 | 0.37 |
| **Structured** | | | | | | |
| vtab/clevr_closest_object_distance | 0.15 | 0.15 | 0.14 | 0.15 | **0.25** | 0.24 |
| vtab/clevr_count_all | 0.22 | 0.15 | 0.22 | 0.21 | 0.27 | **0.29** |
| vtab/dmlab | 0.15 | **0.19** | 0.16 | **0.19** | 0.16 | 0.17 |
| vtab/dsprites_label_orientation | 0.01 | 0.02 | 0.05 | **0.06** | **0.06** | 0.05 |
| vtab/dsprites_label_x_position | 0.03 | 0.03 | **0.06** | **0.06** | **0.06** | **0.06** |
| vtab/dsprites_label_y_position | 0.03 | 0.03 | 0.11 | 0.12 | 0.12 | **0.13** |
| vtab/kitti_closest_vehicle_distance | 0.17 | **0.18** | 0.12 | 0.11 | 0.17 | 0.17 |
| vtab/smallnorb_label_azimuth | **0.06** | **0.06** | 0.05 | **0.06** | 0.05 | **0.06** |
| vtab/smallnorb_label_elevation | 0.11 | 0.12 | 0.12 | 0.12 | 0.12 | **0.13** |

Table B8: **Linear-Probe performance of various CLIP-based models** over 29 datasets using CLIP Benchmark [15]. Scores within the 99.5% Clopper-Pearson confidence interval of each dataset's top score are shown in bold. Baselines contain results for the Standard CLIP [39] and Text2Concept (T2C) [30] models; ImageNet and CC3M column sections contain the B-cosified RN-50 CLIP models trained with cosine and cyclic learning schedulers trained with ImageNet [16] and CC3M [47] datasets, respectively. [47] datasets, respectively. The cyclic learning training inspired from [25]. Dataset type is taken from [15].

| Model | Baselines | | ImageNet [16] | | CC3M [47] | |
| Dataset | Standard [39] | T2C [30] | Cosine | Cyclic | Cosine | Cyclic |
|---|---|---|---|---|---|---|
| **Natural** | | | | | | |
| cars | **0.80** | 0.33 | 0.71 | 0.71 | 0.67 | 0.69 |
| fer2013 | **0.63** | 0.48 | 0.60 | 0.60 | 0.61 | 0.61 |
| fgvc_aircraft | **0.42** | 0.23 | 0.36 | 0.36 | 0.33 | 0.34 |
| gtsrb | **0.84** | 0.69 | 0.82 | 0.83 | 0.81 | 0.82 |
| imagenet1k | 0.71 | **0.73** | 0.72 | 0.72 | 0.67 | 0.68 |
| stl10 | 0.97 | 0.96 | **0.98** | **0.98** | 0.96 | 0.97 |
| voc2007 | 0.82 | 0.82 | **0.83** | **0.83** | 0.81 | 0.81 |
| vtab/caltech101 | **0.92** | 0.88 | **0.92** | **0.92** | 0.86 | 0.86 |
| vtab/cifar100 | 0.70 | 0.70 | **0.74** | **0.74** | 0.71 | 0.72 |
| vtab/cifar10 | 0.89 | 0.89 | **0.91** | **0.91** | 0.88 | 0.89 |
| vtab/dtd | **0.74** | 0.66 | 0.73 | 0.73 | 0.69 | 0.70 |
| vtab/flowers | **0.92** | 0.73 | 0.91 | 0.91 | 0.89 | 0.89 |
| vtab/pets | 0.88 | **0.89** | 0.86 | 0.88 | 0.84 | 0.85 |
| vtab/svhn | 0.65 | 0.60 | **0.66** | **0.66** | 0.63 | 0.65 |
| **Specialized** | | | | | | |
| mnist | **0.98** | 0.97 | 0.97 | 0.97 | **0.98** | 0.97 |
| renderedsst2 | **0.72** | 0.51 | 0.56 | 0.56 | 0.62 | 0.60 |
| vtab/diabetic_retinopathy | **0.76** | 0.75 | **0.76** | **0.76** | 0.75 | **0.76** |
| vtab/eurosat | 0.94 | **0.95** | **0.95** | **0.95** | **0.95** | **0.95** |
| vtab/pcam | 0.82 | **0.84** | 0.82 | **0.84** | 0.82 | 0.81 |
| vtab/resisc45 | **0.91** | 0.84 | 0.89 | 0.89 | 0.87 | 0.88 |
| **Structured** | | | | | | |
| vtab/clevr_closest_object_distance | 0.53 | 0.53 | 0.52 | 0.53 | 0.54 | **0.55** |
| vtab/clevr_count_all | 0.62 | 0.53 | **0.68** | 0.67 | 0.65 | 0.64 |
| vtab/dsprites_label_orientation | 0.61 | 0.49 | 0.57 | 0.58 | 0.61 | **0.63** |
| vtab/dsprites_label_x_position | 0.52 | 0.43 | **0.55** | 0.53 | 0.53 | 0.54 |
| vtab/dsprites_label_y_position | 0.56 | 0.50 | 0.58 | 0.58 | 0.57 | **0.59** |
| vtab/smallnorb_label_azimuth | **0.14** | **0.14** | **0.14** | 0.13 | **0.14** | **0.14** |
| vtab/smallnorb_label_elevation | 0.37 | 0.33 | 0.39 | 0.39 | **0.40** | **0.40** |
| vtab/dmlab | 0.48 | 0.44 | 0.48 | **0.49** | 0.47 | 0.47 |
| vtab/kitti_closest_vehicle_distance | **0.52** | 0.47 | 0.48 | 0.50 | 0.47 | 0.47 |

# C    Implementation Details

We implement our code in Pytorch [35] for all the experiments and use Captum [29] for visualisations.

## C.1    Standard Models

### C.1.1    Models

We B-cosify models from Torchvision [54] supervised on ImageNet [16]. We use a diverse set architectures, including both CNNs (ResNet-18 [22], ResNet-50 [22], and DenseNet-121 [24]), and ViTs [18, 6, 57] with (ViT$_c$-Ti, ViT$_c$-S, ViT$_c$-B, ViT$_c$-L) and without (ViT-Ti, ViT-S, ViT-B, ViT-L) convolutional stems. For ResNet-50, we use both the weights originally released by Torchvision and the updated V2 weights, which constitute models trained for longer and with more augmentations [55].

### C.1.2    Datasets

We use ImageNet [16] to fine-tune all the B-cosified standard models and evaluate them on ImageNet's validation set. For training, we use train transforms - crop size of 224, horizontal flip with 0.5 probability, random resized crop of 224 with bilinear interpolation, Add Inverse transform [10] and modified mean-std normalisation (to accommodate for 6 channel input from the AddInverse). For evaluation, instead of a random resized crop, we do a center crop with a crop size of 224.

### C.1.3    Optimization

For each architecture, we use the B-cosification stategy derived in Sec. 3, and fine-tune for 90 epochs using the AdamW optimizer [26] and cosine scheduling for the learning rate learning rate of $10^{-4}$ for the convolutional models (since the standard pre-trained models end with a learning rate of $10^{-4}$ at the 90$^{th}$ epoch, from which we want to fine-tune further). For ViTs, as the learning rate decays to a very small value, we tested with different learning rates ($10^{-3}$, $10^{-4}$, $10^{-5}$) and found $10^{-3}$ worked best for all the models. Also, we only use a linear learning rate warmup of 10,000 steps with a decay of 0.01 for the base and large ViT models.

### C.1.4    Experiments

**Increasing B:** We tested three different setups for increasing B. 1) Discrete B setting to $1, 1.25, 1.5, 1.75, 2, 2.5, 3, 5, 7$; 2) Linear increase of B in $n$ epochs from B=1 to B=2. We used $n = 5, 10, 20, 45, 90$; 3) Learning B parameter to increase to B=2 using weight decay with coefficients $0.2, 0.5$ and $0.9$. See Tab. 2 for results.

**Removing biases:** We test two setups for removing the biases from the network: 1) Removing all the bias parameters; 2) Decay the bias parameter using the weight decay with coefficients $0.5$ and $0.9$. See Tab. 3 for results.

**Impact of pre-trained weights:** To check the impact of pre-trained weights on fine-tuning, we fine-tuned weights from CLIP [39] ResNet-50 [22] , DINO ResNet-50 [12], and Torchvision [54] ResNet-50 weights v1 and v2 (long trained recipe) [55].

### C.1.5    Evaluation

As in Sec. 3.2, we evaluate both for classification accuracy and for interpretability using the GridPG [9] metric. We compare both accuracy and interpretability of the B-cosified models with B-cos models trained from scratch from [11]. For interpretability, we also compare with several post-hoc attribution methods as baselines, namely Guided Backprop [51], Gradient [49], DeepLIFT [48], IxG [48], IntGrad [53], and GradCAM [46]. Qualitatively, we visualize the colored B-cos explanations and the attribution maps [11].

## C.2    CLIP Models

We use a CLIP [39] ResNet-50 [22] model for B-cosification.

### C.2.1 Datasets

We use ImageNet [16] and CC3M [47] to fine-tune all the B-cosified CLIP models and test them on multiple datasets from CLIP benchmark [15]. For training, we use the same transform setup as the standard B-cosified models. We use train transforms - crop size of 224, horizontal flip with 0.5 probability, random resized crop of 224 with bilinear interpolation, and Add Inverse transform [10] and modified mean-std normalisation (to accommodate for 6 channel input from the AddInverse) as discussed in the paper. For evaluation, instead of a random resized crop, we do a center crop with a crop size of 224.

### C.2.2 Evaluation

We use the CLIP benchmark [15] for zeroshot and linear probing experiments with the default parameters provided in the official benchmarking code. For text-based localisations, we use the text-based templates from the CLIP for the ImageNet dataset and use them to encode the text features. As text encoder, we use the CLIP ResNet-50 text encoder. The cosine scores between the B-cosified CLIP's image encoding and the pre-trained text encoder are used to do B-cos style localisations and calculate the GridPG scores. We use the unpooled features technique at inference to increase the localisation focus.

We use B-cosified CLIP ResNet-50 fine-tuned on ImageNet using SigLIP loss [59] and cosine scheduling for visualisation.

### C.2.3 Optimization

We use the Adam optimizer [26] and fine-tuned models till 90 epochs, while the CC3M models are fine-tuned for 30 epochs. The size of CC3M is approximately three times that of ImageNet, so the trained models are comparable. Keeping consistent with the Standard B-cosification recipe, we train with a learning rate of 1e-4 using cosine scheduling. SigLIP contrastive loss [59] is used to train the models.

