# OpenReview forum: "B-cosification: Transforming Deep Neural Networks to be Inherently Interpretable"
_NeurIPS.cc/2024/Conference — NeurIPS 2024 poster_

### Official Review · Reviewer_JC7z · 2024-06-28

**Soundness:** 3
**Presentation:** 3
**Contribution:** 3
**Rating:** 7
**Confidence:** 4

**Summary:**

This work proposes B-cosification, a method to transform a pre-trained network to a B-cos network [1,2]. Consequently, the transformed model can be finetuned for better explanations of the model behavior while retaining predictive performance. Experiments are done on various CNN and Transformer architectures, as well as a case study on CLIP.


[1] Böhle, Moritz, Mario Fritz, and Bernt Schiele. "B-cos networks: Alignment is all we need for interpretability." Proceedings of the IEEE/CVF Conference on Computer Vision and Pattern Recognition. 2022.

[2] Böhle, Moritz, et al. "B-cos Alignment for Inherently Interpretable CNNs and Vision Transformers." IEEE Transactions on Pattern Analysis and Machine Intelligence (2024).

**Strengths:**

* B-cos networks are an upcoming, interesting research direction. The need to train a network from scratch is commonly considered a limitation (e.g., see the Concept Bottleneck Model Literature [3,4]); therefore, alleviating this need is a valuable contribution to the research landscape.
* The paper is clearly structured and well-written. Only towards the end of experiments, it becomes a bit cramped with information, which is minor.
* Experiments are performed over a broad range of backbones. Also, I appreciate that the authors make the effort to actually measure the interpretability rather than just stating that it is interpretable.

[3] Yuksekgonul, Mert, Maggie Wang, and James Zou. "Post-hoc Concept Bottleneck Models." The Eleventh International Conference on Learning Representations.

[4] Marcinkevičs, Ričards, et al. "Beyond Concept Bottleneck Models: How to Make Black Boxes Intervenable?." arXiv preprint arXiv:2401.13544 (2024).

**Weaknesses:**

* My biggest uncertainty is about the stated (inherent) interpretability. Firstly, to me, I see B-cos(ified) networks more as a post-hoc gradient-based explanation method, which additionally regularizes during training for more faithful explanations. However, there is no certainty upon seeing an explanation that the model actually acted upon these highlighted pixels, which to me makes B-cos networks not inherently interpretable. In other words, a regularizer does not enforce that the model truly depends on the given explanations, and therefore, the explanation given is not necessarily faithful to the underlying model. Thus, I also disagree with the Abstract that states that the explanations are "model-faithful by design".
* Followingly, such a method should be evaluated with respect to faithfulness to the underlying model. As such, I would be very interested in how B-cosified networks perform in some faithfulness sanity checks such as the Model Parameter Randomization Test of [5].
* Without any standard deviation, it is hard to assess the (statistical) significance of performance improvements. Also, multiple seeds (up to some degree) prevent the all-too-common hyperparameter optimization on the test set.
* The CLIP explanations when B-cosifying CLIP are on par with GradCAM on the pre-trained weights. To be better requires the introduction of a new, arbitrary aggregation of value vectors.
* If I understand it correctly, this new aggregation would also affect CLIP's performance. It is not explored how the model performance on CLIP Benchmark would be affected by this aggregation that appears to make CLIP more explainable.
* While this work has high significance in today's world of foundation models, the methodological novelty is limited, as most B-cosification tricks have already been introduced in [2].
* In my opinion. speedups of 2x are not necessarily big enough to warrant that it's a huge improvement over training from scratch.
* Page 1 appears to be missing some vspace after line 37.

[5] Adebayo, Julius, et al. "Sanity checks for saliency maps." Advances in neural information processing systems 31 (2018).

**Questions:**

* In line 295, is "interpolate" really the right word (it confused me slightly in my read). That is, if it was interpolating, then shouldn't Fig. 5a B-cos FeatureCLIP for p=1 be equal to B-cos CLIP? (Also, there are some minor reference naming issues with respect to the figure in line 292-297, and the legend of Fig. 5a needs to be updated)
* While the Grid Pointing Game appears to be a good metric to measure how well the explanations are localized, it is not telling enough on a fine-grained level how informative the explanations are. While the qualitative figures 1 & 5b give some insights, into this, they're obviously cherry-picked (which I don't condone). What I like in papers with qualitative examples is that, in the Appendix, there are also some randomly chosen examples so that an interested reader can get a feeling for how the method actually performs. Thus, I would be interested in seeing some more qualitative examples that are (truly) randomly drawn.
* How well would the proposed method be transferrable to different input modalities such as text?

**Limitations:**

One could maybe write one more sentence with respect to which models are *not* B-cosifiable. Apart from that, limitations are adequately discussed.

---

> ### Author Rebuttal · Authors · 2024-08-06
>
> Thank you for your detailed review and valuable feedback. We appreciate your positive remarks on our work's significance and your constructive suggestions for improvement. We have carefully considered your comments and would like to address each point you raised.
> - **Inherent Interpretability of B-cos:** Please note that B-cos explanations are not mere post-hoc explanations. Instead, the linear summaries W(x) of the models are in fact exact summaries (i.e., yielding the same outputs when applied to x) of the models’ computations and are optimized during training to align with task-relevant features, as described in [6], and have further been found to compare favorably to post-hoc explanations (Figure 6 in [6]). While we agree that the B-cos explanations of course constitute a simplification of the full network, previously reported results suggest that they indeed capture important aspects of the models’ computations, allowing them to e.g. localise class-specific features at a fine granularity. Our work builds on those prior findings and examines how to bring these previously reported benefits to pretrained conventional models.
> - **Model Randomization test performance:** Note that the localisation scores give clear evidence that the explanations are highly model-dependent: if they were not, they would not allow for localizing specific classes with such high specificity (>90%, see Figure 6 in [6]). Further note that conversely, explanation methods that failed the randomisation tests, as proposed in Adebayo et al., notably score very low in the grid pointing game. We are happy to include results regarding the model randomisation tests in the final version of the paper. Lastly, as can be seen in the new figure we provide in the attached global response PDF (Figure 6, right), before fine-tuning the models for inherent interpretability, they give highly uninterpretable explanations (cf. ‘at start’).
> - **Multiple seeds:** As we write in our global response, we now provide results over three runs corresponding to Tables 2, 3, and 4 in Tables 5, 6, and 7, respectively, with standard deviations. From tables 5-6, we find that the conclusions in Section 3.2.2 remain unchanged — the simplest approach of directly setting B=2 and removing the bias at the beginning performs as well as more involved strategies such as learning B or decaying bias. Also, from Table 7, we find that the results regarding classification and interpretability performance, as in Section 4.1, are supported by repeated experiments.
> - **Aggregation of CLIP vectors:** We agree with the reviewer that the aggregation mechanism changes the model’s behavior and thus does not faithfully reflect the full model’s predictions. We introduce this aggregation mechanism to show that our B-cosification approach allows us to interpret intermediate features of the CLIP model at a high level of detail (clearly outperforming the localisation ability of both full models), as the features still maintain a higher level of spatial resolution. This highlights an additional aspect in which B-cosified models could help better understand the inner workings of foundational models.
> - **Speedups in fine-tuning:** We find that the speedups are often much better than 2x for convolutional models (Table 4). We agree that the speedups are lower for ViTs, and believe this could be because our method could further be optimized for ViTs, e.g. see the discussion on ViT performance in response to reviewer DdYE.
> - **Interpolation:** For aggregating and explaining the intermediate features of the CLIP model, many options exist - e.g., attention pooling as in the original (and B-cosified) CLIP model, average pooling, or max pooling. To better understand the implications of the feature aggregation, we propose a mechanism based on the cosine similarities that allow us to interpolate between average (p=1) and max pooling (p=inf); note that since both of them differ from attention pool (denoted by B-cos CLIP in the figure), neither of them give the same results as attention pool. These results also indicate that the attention pooling mechanism selects the embedding features in a more specific fashion than mere average pooling, but aggregates the features in a coarser fashion than selecting only the features with high cosine similarities (p$\gg$1).
> - **Qualitative examples:** We provide additional randomly selected examples corresponding to both Figures 1 and 5b in Figure 7 of the attached global rebuttal PDF and will provide an extended set as a supplement to the final version of the paper.
> - **Performance on text modalities:** We agree this would be a very interesting investigation. Since B-cos models were primarily used for image classification, we also focused on the same setting in this work. However, exploring how to extend to other modalities, such as text, would be a fruitful direction for future research.
> - **Which models are not B-cosifiable?** Based on our experimental results, we believe this question does not have a yes or no answer - different models might benefit from a B-cosification to different degrees. For example, while the convolutional models we tested adhere closely to the original formulation of the B-cos models and thus lend themselves well to B-cosification with a high degree of localisation, our results regarding the feature aggregation mechanism (attention pooling vs. our proposed approach) suggest that some models might lack the necessary alignment pressure to exhibit equally strong localisation abilities. We appreciate the reviewer’s suggestion and will update our discussion accordingly.

---

> > ### Comment · Reviewer_JC7z · 2024-08-09
> >
> > I thank the authors for their response and especially appreciate their candor in presenting Fig.7.
> > I have increased my score. While the methods proposed are not the most novel, this work might pave the way for a broader applicability of B-cos-like networks, which I deem desirable.

---

> ### Comment · Reviewer_JC7z · 2025-01-25
> **Publishing of code**
>
> I would like to remind the authors to publish their code.
>
> The answer in the paper checklist says "... and [we] will make all our modifications available for ensuring full reproducibility of the reported results." Yet, the github repo https://github.com/shrebox/B-cosification remains empty.
>
> My score was influenced by this belief, as I believe the additional usability of B-cos is a major positive point of this work.
> As such, I encourage the authors to adhere to the statements made during the submission period in a timely manner. Also, it's bad for distributing your work, as other (like me) won't use this work if there is no codebase available.
> Best regards

---

### Official Review · Reviewer_DdYE · 2024-07-11

**Soundness:** 4
**Presentation:** 4
**Contribution:** 3
**Rating:** 7
**Confidence:** 4

**Summary:**

This work builds on the recent line of works aiming to design inherently interpretable models. In particular, it considers the recently proposed B-cos networks and shows how pre-trained conventional CNN/VIT models can be converted into a B-cos network through fine-tuning. The authors first discuss which parts of conventional DNNs can be converted into an equivalent B-cos model and then propose two simple mechanisms that address the two missing components (increasing B and removing the biases) through fine-tuning. Results illustrate that the proposed B-cosified models are able to obtain equivalent accuracy to both the original model as well as the standard B-cos network, while obtaining similar localization results to the B-cos network in a shorter time due to the re-use of the original weights.

**Strengths:**

The work addresses an important problem of converting pre-trained black-box models into inherently interpretable models. Instead of requiring complete retraining, which often restricts the applicability of these models, only requiring fine-tuning is a first step into the right direction.

The paper is well structured and the different modifications required are presented in sufficient detail.

The experimental evaluation on both CNN and Transformers as well the case study on CLIP shows the potential of the proposed approach.

**Weaknesses:**

It would be beneficial to include a plot on how accuracy and localization score vary as fine-tuning progresses for one/some of the standard models (CNN/VIT).

In Sec. 3.2.2, it is unclear how the bias is handled when performing the ablation study in Table 2. Also, the B-cosified model consistently outperforms the baseline in Table 2 and 3, do the authors have an hypothesis on why this fine-tuning process increases performance?

Minor:

It would be beneficial to also provide the number of fine-tuning epochs that were required in Table 4.

Figure reference missing in Line 141.

**Questions:**

For the ViT results, there are some more cases where the B-cosified networks performance degrades slightly compared to the black-box and the localization is slightly lower than for the B-cos model. While these differences are small, do the authors have a hypothesis on their reason? Are these induced by the interpretation of the GeLU?

**Limitations:**

The limitations of the proposed method have been briefly but sufficiently discussed.

---

> ### Author Rebuttal · Authors · 2024-08-06
>
> Thank you for your valuable feedback. We appreciate your recognition of the significance of our work and your constructive suggestions. We have carefully considered your comments and would like to address each of the points you raised.
> - **How do localization scores vary with epochs?:** As we write in our global response, we now report these results in Figure 6 in the attached PDF. We find both quantitatively (left) and qualitatively (right) that the localization improves very quickly. For example, for ResNet-18, the localization improves from 21.5 to 82.0 after just one epoch of fine-tuning, which is close to 86.9 achieved by the B-cos model trained from scratch. On the other hand, modifications to the model architecture at the beginning of the B-cosification process lead to an initial drop in accuracy, from 69.8 for the standard model to 59.9 after one epoch. However, we recover the accuracy much quicker than training from scratch, with the B-cosified model needing just 29 epochs to reach the accuracy of a B-cos model trained from scratch for 90 epochs.
> - **How is bias handled?** Apologies if this was not sufficiently clear. In the ablation studies, we remove the bias when varying B since that is the setup in B-cos as well as our B-cosification process (Table 3); for an additional discussion, also see ‘Completeness/bias’ in answer to reviewer Z7HS.
> - **Performance of B-cosified models in Tables 2-3:** The baseline in Tables 2-3 refers to the standard and B-cos models trained for 90 epochs. During B-cosification, we start from the 90 epoch trained standard model and further fine-tune, which leads to performance gains. For a fairer comparison, we also report results when fine-tuning the baselines for an additional 90 epochs in Table 8 (blocks 3-4) in the global response PDF, and find that the B-cosified models perform similarly or better, while also being interpretable.
> - **Speedup epochs:** In Table 4, we always fine tune for 90 epochs, however, the speedup column shows the point at which the B-cosified model reaches the performance of the corresponding B-cos model when trained from scratch. We now also report the epoch number at which this ‘overtake’ happens at in Table 8 ($t_\text{ovt}$) in global response PDF, e.g. for ResNet-18, we find that the B-cosified model reaches B-cos performance with just 29 epochs of fine-tuning.
> - **ViT performance:** Please note that in comparison to many other post-hoc explanation methods, the localisation scores of both B-cos and B-cosified ViTs are relatively high, differing only by a few percentage points. We will analyse and discuss (e.g., impact of GELU)  potential reasons for the differences between the ViT models in more detail in the final version.
> - **Missing reference:** Thank you for pointing this out, we will correct the missing reference for the final version.

---

> > ### Comment · Reviewer_DdYE · 2024-08-12
> > **Reply to Author Response**
> >
> > I would like to thank the author for providing the additional clarifications as well as the additional results on how the localization scores change during fine-tuning. After reading the rebuttal as well as the other reviewer's comments, I will retain my 'accept’ score and believe that this will make a valuable contribution to the conference.

---

### Official Review · Reviewer_71N9 · 2024-07-12

**Soundness:** 3
**Presentation:** 3
**Contribution:** 2
**Rating:** 6
**Confidence:** 4

**Summary:**

This work targets the interpretability of modern architectures via a process that the authors call B-cosification. Contrary to the original B-cos networks that are trained from scratch by architecturally enforcing alignment between inputs and weights, B-cosification constitutes a post-hoc method, aiming to convert/finetune exististing pre-trained models towards interpretability.

**Strengths:**

This work aims to transform existing pre-trained models to "inherently interpretable" ones via a post-hoc b-cosification method. Contrary to other post-hoc approaches, this finetuning allows for essentially altering the properties of the network, bypassing some of the criticisms of post-hoc approaches.

Overall, this work constitutes an interesting investigation of how to transform standard architectures to B-cos like ones.

**Weaknesses:**

Even though this is an interesting approach, its novelty is limited. The authors explore how to finetune the model to match the b-cos alignment pressure, by exploring some architectural changes like normalisation and bias decrease.

One of the main issues the proposed process is the complexity of the finetuning. Some of the ResNet architectures considered in the main text are trained for around 90 epochs, e.g., on ImageNet. That is a common number of epochs to train these models from scratch. Can the authors provide a summary/visualisation of the acc/localization with respect to the number of epochs?

What is the behaviour of the base models when finetuned the same way as in the proposed b-cosification process (without b-cos)?

Can the authors add another column to Table 4, denoting the accuracy of the standard trained from scratch b-cos networks?

The text needs to be proofread. There are several sentences that lack clarity, typos and other issues, e.g., line 141 "see also Fig. XX, whereas conventional models use 3 channels".

The authors are encouraged to remove the spacing altercations that do not follow the format of the conference.

**Questions:**

Please see the Weaknesses section.

**Limitations:**

The authors have a limitation paragraph in the main text.

---

> ### Author Rebuttal · Authors · 2024-08-07
>
> Thank you for your constructive feedback on our work. We have carefully considered your comments and would like to address each point you raised.
> - **Limited novelty**: To the best of our knowledge, we are the first to investigate how to transform existing uninterpretable models into inherently interpretable B-cos models. In particular, we find that we can effectively B-cosify existing models at a fraction of the cost of training them from scratch, which has the potential to make interpretable models far more accessible than they currently are. In order to be able to do this, we discuss in detail how the models differ and devise specific solutions to address those differences. In particular, we first show that it is possible to cast conventional models in the same framework as B-cos models without changing the function that they represent (i.e., we adapt the model $f_{org}$ to a B-cos version $f_{Bcos}$ with B=1 whilst maintaining functional equivalence, such that $f_{org}(x) = f_{Bcos}(x) $ $\forall$ $x$ ). Specifically,  in this context we address the input encoding and normalization, activation functions, weight normalization, and incorporate the pre-trained weights. Thereafter, to increase interpretability, we increase the parameter B and decrease the models’ biases. This, of course, functionally changes the model and hence requires fine-tuning as a final step to adapt and recover the model’s performance to the new changes. We are unaware of previous work that explored this and would be grateful if the reviewer could point us to relevant related work that would limit the novelty of our contribution.
> - **Complexity of fine-tuning**: We indeed train the models for a commonly used number of epochs (90) - however, to understand the efficiency gains by using pre-trained models, apart from accuracy and localisation scores, we also report the speed-up for reaching the same accuracy as randomly initialized B-cos models in the results (Table 4). We are very grateful for the reviewer’s suggestion to provide a visualization of accuracy and localisation scores over the training epochs, which we added in the attached PDF  in Figure 6, as this helps clearly see how quickly the models’ explanations improve in their localisation (**localisation scores > 80% after the first epoch**). While the accuracy generally improves over the course of 90 epochs, we find that they tend to reach the performance of B-cos models trained from scratch very quickly (e.g. **ResNet-18 reaches the performance of the original B-cos models within 29 epochs**). This is our primary contribution.
> - **Fine-tuning base models**: In Table 8 (column blocks 3-4) in the PDF attached, we report the results for the standard pre-trained and B-cos pre-trained models fine-tuned further for 90 epochs in the same way as proposed in the B-cosification process for two CNNs and two ViTs models. We find that even after fine-tuning the pre-trained models (standard and B-cos) further, B-cosified models perform competitively with conventional DNNs (column block 3) and outperform the B-cos DNNs (column block 4) consistent with the original findings mentioned in the main submission. We will provide full results in our revision.
> - **Accuracy of B-cos trained from scratch**: In Table 8 (column 5) in the global response PDF, we now report this additional column for two CNNs and two ViTs models. We will provide full results in our revision.
> - **Writing fixes**: Thank you for pointing these out; we will make the changes in our revision.

---

> > ### Comment · Reviewer_71N9 · 2024-08-13
> >
> > I would like to thank the authors for their thorough rebuttal to all the reviewers' questions. Having carefully read the clarifications and the other comments, I decided to increase my score.

---

### Official Review · Reviewer_Z7HS · 2024-07-12

**Soundness:** 3
**Presentation:** 2
**Contribution:** 3
**Rating:** 7
**Confidence:** 4

**Summary:**

The authors discuss the B-cosification of a pre-trained model.
The B-cosification involves changing the operations performed in the linear layers to one involving a cosine. Not all properties of B-cos models are in the end satisfied, yet performance in terms of accuracy and localization are on par with a fully trained B-cos model.

**Strengths:**

- The method contributes to the current effort to turn pre-trained models into explainable ones without full retraining.
- The performance dropped are limited both against the original model and the equivalent B-cos model

**Weaknesses:**

- The method still requires a full fine-tuning of the weights.
- The experimental setting is not very clear.
- The presentation is not always clear and sometimes verbose.

**Questions:**

* P1 L33: which are increasingly popular—can cost millions of dollars.
    What if I am not using dollars? Maybe time would be a better metric? Or energy?

* P2 L38: the recently proposed B-cos Networks.
    Until here, B-cos was never defined. Since it is not a contribution of this paper, would you have a citation to guide the reader?

* P2 L42: What does "functionally equivalent" mean? What do you mean with "alignment pressure"?
Maybe instead of introducing these vague notions, why not use the opportunity given by this paragraph to explicitly say what B-cos consists of, namely (up to my understanding) replacing the linear operations of the hidden layers by a cosine-based function parametered with B? You can go further and say that for B=1, the operation is ("is" and not "is equivalent to") the classic linear matrix multiplication. "Linear" implies the absence of a bias term. Otherwise, that would be the "classic affine matrix multiplication". Nevertheless, one can stress this further.

* P2 L45: "significantly more interpretable explanations (Fig 1)"
    What do you mean by "more interpretable explanations"? What I see in the third row of Fig1 is kind of a mix between the second row and an edge detector or an Integrated Gradient saliency map. Is capturing the shape or details of the "important part of the image" making an explanation more interpretable?
    Following this discussion, the result of GradCam+CLIP on the first example is preferable from my point of view because one can see that the model uses the help of the grass to actually predict a sheep. The same applies to the "[...] photo of a boat."

* P2 L47: "On supervised settings, we find that B-cosified models often outperform"
    How is performance measured here? Classification accuracy or caption prediction?

* P2 L53-57: I am a bit confused when you say "design[ing] inherently interpretable models". I do not understand your contribution as a new design but a conversion. This is a valuable and efficient contribution that aligns with current concerns, be it ecological, time efficient, or simply democratizing ML for the majority who do not have access to huge computational resources.
    This work fits into the recent line of research investigating how to alter the architecture of trained models to make them "interpretable." Why not stress this aspect more?

* P2 L65-: I am unsure if the claimed results are that important to count as contributions.
    Again, what you mean by "significantly interpretable" is not clear. Could you please clarify?

* Sec2: If you choose to attach your contribution to the line of "converting trained models" (you don't have to) to "interpretable ones", I suggest some works:
    * Stalder, S., Perraudin, N., Achanta, R., Perez-Cruz, F., & Volpi, M. (2022). What you see is what you classify: Black box attributions. Advances in Neural Information Processing Systems, 35, 84-94.
    * Aytekin, C. (2022). Neural networks are decision trees. arXiv preprint arXiv:2210.05189.
    * Gautam, S., Boubekki, A., Höhne, M. M., & Kampffmeyer, M. (2024) Prototypical Self-Explainable Models Without Re-training. Transactions on Machine Learning Research.

* P3 L111: "Many common DNNs consist of a series of blocks of linear layers followed by non-linear ReLU activations [29], and are thus piece-wise linear functions1 : i.e., for every input x, they effectively compute a linear transformation of that input: y(x) = W(x)x + b(x)"
    What are W and b? I assume the model's input and output are x and y, respetively.
    If it is "effectively [...] a linear transformation", why are W and b functions of x?

* P3 L113: "dynamic linear"
    Why do you need to introduce this notion? What is dynamic in an affine matrix multiplication?

* P3 L116: "This linear mapping W(x)". The same goes here: W cannot be a linear mapping/function if we are talking about "a piece-wise linear model".

* P3 L120: Why does a "complete explanation" requires "y(x)=W(x)x"? Two sentences later, you actually contradict this claim.
    I fail to grasp why having a bias is so problematic if most of the backward operations are based on the gradient. Could you clarify?

* P4 L123: Can you clearly define "alignment pressure" and why it is necessary to be introduced?
    From this paragraph, I understand that the conversion from line (B=1) to (B>1)-cos model has already been investigated in [6] and [7]. Has it?
    If it has been, this jeopardizes the relevance of your main contribution.

* P4 Eq1: How do you compute $cos(x,w)$?
    I understand that $x$ is a vector, $W$ is a matrix, and $\times$ is the element-wise multiplication.

* P4 L127: Why do we need the notion of "dynamic linear"?
    If it has to do with W being a function of x: if B=1, W does not depend on x.

* P4 L141: Missing reference with Fix XX.

* P4 L146: "In particular, we show that a conventional model can be framed as a functionally equivalent B-cos model as in [7] with B =1, which additionally employs bias terms. Only upon modifying these two aspects, i.e., biases and B, does the model need to be fine-tuned to adapt the weights to those changes."
    These two sentences are not clear.

* P4 L150: "As mentioned in Sec. 3.1, B-cos models use input representations with six color channels"
    This was not mentioned in Sec 3.1. This seems to be an atavism from an older version of the paper. Please fix.

* P4 "Input Encoding and Normalisation": The operation is straightforward: B-cos models requires 6 channels: so inputs need to be transformed according to Eq2 and the first layer of the B-cosified models is discarded and replaced by one with 6 channels (implying a full training thereof).

* P5 "Activation Functions." You start the section by saying that activations are unnecessary, yet you introduce them. Why?

* P5 "...unit norm weights, ..., which the authors motivated by the fact that the only way any given neuron can achieve its maximal output is by increasing the weight-input alignment, which in turns[typo] leads to the improvements of the explanations."
    What is "weight-input alignment"? I fail to see why normalized weights improves the explanations.

You could streamline this section by leaving the justification of the B-cos meaningfulness to an introductory paragraph and focusing on the changes required by your approach. There is no need to justify or make claims about the benefit of B-cos since it is not one of your contributions here.

* P5 Section 3.2.2
    None of the tables present statistical tests, which makes it difficult to draw any conclusion, especially given that the results are quite similar from one setting to another.
    * How many times the experiments were repeated? What are the standard deviation? Which result is significantly better?
    * What is the localization score?


* P6 L204: "setting B = 2 and then fine-tuning, yields performance that is on par with
learnable B parameters,"
    If this holds, it is also on par with Linear B 90 epochs.

* P6 Table 4: The increments are quite small. Have you considered running a t-test to identify significant differences?
    Is the "speedup" with respect to the fully trained B-cos model? What about the "speedup" with respect to the vanilla models with classic matrix multiplication?

* P7 L245: "Specifically, we find that averaged across architectures, B-cosified models outperform B-cos models trained from scratch by 2.31 pp, with an average training speedup (to match performance) of 2.96x."
    This is an important result. Why not highlight it in a separate section/ablation study?

* P8 Interpretability: In the text you refer to paper [6] while in the caption of Fig2 you refer to [7] (These papers share essentially the same figures). Fig2 would be better positioned closer to its text.
    Finally, interpretability is defined in paper [6] (or rather in paper [6] of paper [6]) as the performance at the "grid pointing game". If I understand it, it is related to the sum of the GCAM attribution to each class of the grid cell. My understanding of gradient-based explainable methods is that the scale of the output saliency maps is inconsistent. Is it valid? If yes, how does it affect the score returned by the Pointing Game? How robust is this score against possible spurious relationships between grid cells? How does performing well in this game relate to the interpretability of the model's explanations?

* P8 Fig4: What are Natural/Specialized/Structured Data?

* P9 L284: "We find that the B-cosified models significantly outperform the Text2Concept approach and achieve accuracies that are more similar to the original CLIP’s zeroshot and linear probing accuracies."
    Text2Concept is a quite different approach. In the original paper thereof, they report performance close to CLIP's. How do you explain this difference? What is different in your setting? Do you train supervised or self-supervised?

* Evaluating Model Interpretability.
    In the image classification case, you compute the explanation based on a single label; the equivalent case here is thus to back-propagate using the
    Isn't the average similarity equivalent to the similarity with an out-of-distribution point in the text embedding?
    Why did you choose p=7 and p=19?

    Overall I did not quite understand this section. I am not very familiar with CLIP, so it is not easy for me to follow what is happening.

---

> ### Author Rebuttal · Authors · 2024-08-07
>
> Thank you for the detailed feedback and the suggestions to improve clarity; we will incorporate them as well as the following answers in our revision.
> - **Full fine-tuning**: The research question we examine is how to leverage existing pre-trained uninterpretable models to make training more interpretable ones more cost-effective. To endow pre-trained models with _inherent interpretability_, it is indeed inherently necessary to fine-tune the full models, as we aim to fundamentally change how they operate - however, we find that this approach is significantly cheaper than training the corresponding interpretable models from scratch (up to 9 times speed-up, Tab. 4).
> - **Multiple runs**: See global response.
> - **Tab 4 small increments**: The increments in accuracy are indeed small but also not our core focus. Instead, we show that it is possible to _maintain the original models’ accuracy_ whilst significantly increasing their interpretability, at a significantly lower cost than when training from scratch as in [6].
> - **B-cos more interpretable**: As reported in [6,7], B-cos explanations **explain the full model from input to output** and are significantly more localized, and allow to visualize the important features in color (Figs. 1,7 in [6]), while also allowing to understand misclassifications (Fig 9, [6]), making it easier for humans to see understand the model's predictions. Please note that in the examples in Fig. 1, the rows correspond to different models, and so are not directly comparable. I.e., the CLIP model might indeed use the grass to predict the sheep, while the B-cosified CLIP model might not.
> - **Comparing to Text2Concept**: [28] shows that a linear transform can map an arbitrary model's features to CLIP features. Hence, training such a transform on an existing B-cos model serves as a good baseline for B-cosification. We follow [28] and use ImageNet to learn the linear map. However, other parameters are different: we use a B-cos encoder instead of a standard model, a CLIP ResNet-50 instead of ViT-B/16, and evaluate across a broader range of datasets (Fig. 4). Understanding the performance difference requires further investigation, but is orthogonal to our current work.
> - **Dynamic linearity**: As discussed in prior work (cf. [18]),  a linear layer followed by ReLU or MaxOut is  ‘piece-wise linear’: for every input x, the prediction is effectively computed via a linear transformation of x, with the choice of the linear matrix W depending on x. Dynamic linearity (cf. [6, 7]) is a generalization of this concept, making the dependence of the linear matrices on the input more explicit. In particular, ‘dynamic linear models’ compute their output via an input-dependent linear transformation of the input y(x) = W(x) x. Finally, we agree with the reviewer that when bias terms are employed, “piecewise affine” is more accurate in contrast to commonly used nomenclature; given that the bias can be modeled via an additional input dimension, however, we opted to stick to standard nomenclature.
> - **Completeness/bias**: The importance of biases has been discussed in prior work: [48] discusses how gradient-based explanations tend to neglect important contributions from biases (cf Eq. 1, [48]), which have been shown to play a significant role in the models’ outputs (Fig. 13 in [7]); completeness further constitutes one of the foundational axioms of IntGrad [49]. We will revise to better reflect these connections.
> - **Novelty of conversion**: No, while [6] discussed and emphasized the differences between B-cos and conventional networks, we leverage the similarities and show that the existing weights of pre-trained DNNs can be used to efficiently obtain performant and interpretable DNNs, despite e.g. using a different number of input channels (3 vs. 6) and activation functions.
> - **Input encoding**: No, we do not discard the first convolutional layer and train a new one from scratch; instead, as part of our contributions, we show how to construct a first layer and compute the corresponding weight that takes six channels as input and is nonetheless functionally equivalent to the original layer **with no additional training**.
> - **Activation functions**: We keep ReLU since (i) we want to at first maintain functional equivalence with the original model, (ii) using ReLU does not affect the dynamic linearity of B-cos, (iii) the pre-trained weights were trained with them, so using ReLU keeps us closer to the original models.
> - **Normalized weights**: Using normalized weights ensures that the output can be maximized only if the input and weight vectors align, i.e. if the cosine term in the dot product between them is high, see [6]. This helps interpretability, since weights being aligned to inputs better highlights the most important regions.
> - **Localization score**: It refers to the GridPG metric used in [6,7].
> - **Robustness of GridPG**: GridPG [6, 7] computes the ratio of attributions in a grid cell to the total attributions, and so is invariant to scaling To reduce the risk of spurious relationships, the metric uses grids where each image is of a distinct class and is classified correctly with high confidence by the original model, implying that it is unambiguous to the model.
> - **CLIP Interpretability**: We show p=7 and p=19 for illustrative purposes, the full trend across p is shown in Fig 5a. We also provide a more detailed discussion on the CLIP experiments in our response to reviewer JC7z.
> - **Natural/Specialized/Structured data**: We follow the CLIP benchmark [26] for zero-shot and linear probing results (Fig. 4) and categorize datasets based on the images they contain.
> - Sec 3.1: We do mention it, see L139.
> - **"Outperform" performance measure**: We report accuracy for image classification, and zero-shot performance for CLIP models.
> - **Citations**: Thank you, we will add them.
> - **Eq1**: We use row-wise dot products, following prior work (cf. Eq. 9 in [6]). We will revise for clarity.

---

> > ### Comment · Reviewer_Z7HS · 2024-08-12
> >
> > I would like to thank  the authors for their detailed answer.
> >
> > I still have two comments:
> >
> > - *Dynamic linearity* Your explanations comfort me in my understanding that it means "piece-wise linear". So I still don't understand why we need a new word to say that.
> > - *Normalized weights* I failed to find where the advantage of it is studied in [6], except in the limitation because of the computation overhead, and the visualization paragraph.
> > To maximize a cos, only the angle is important. Not the norm (I guess that's why $w$ doesn't have a hat in the cos of Eq3). the matrix $W$ appears in Eq3 outside the cos, so the norms of its rows influence the norm of the output. Again I don't see why it is problem, or where is the advantage. If it is just for visualization, I wonder if normalizing before visualization is not enough?
> >
> > I follow my fellow reviewers and increase my score to accept.

---

> > > ### Author Response · Authors · 2024-08-13
> > >
> > > Thank you for your comments. For the remaining questions, we elaborate below:
> > >
> > > **Dynamic Linearity:** Apologies for not making it sufficiently clear, we will make sure to better clarify in the final paper. Dynamic linearity and piece-wise linearity are not the same. In particular, dynamic linearity describes the general notion of linearly transforming the input with an input-dependent matrix $ \mathbf W(x)$, whereas piece-wise linearity refers to one particular case of dynamic linearity. E.g., in B-cos transformations, the rows of a fixed matrix $\mathbf W_{static}$ with trainable parameters are scaled by an additional cosine factor (compare Eq. 11 in [6]) to give $ \mathbf W_i( \mathbf x) = \cos( \mathbf W_i,  \mathbf x) \times  \mathbf W_{static, i}$ for row $i$. In contrast, in piece-wise linear ReLU networks, $ \mathbf W_i( \mathbf x)$ is given as $ \mathbf W_i( \mathbf x) = [\mathbf {0}$ if  $\cos( \mathbf x,  \mathbf W_{i, static}) < 0$ else $ \mathbf W_i ]$. While both of them are dynamic linear, only the latter is piece-wise linear (i.e., the function consists of **two linear pieces**, one for $\cos( \mathbf x,  \mathbf W_{i, static}) < 0$ and one for $\cos( \mathbf x,  \mathbf W_{i, static}) \geq 0$).
> > >
> > > Following [6], we believe this to be a useful distinction since for any dynamic linear model, $\mathbf W(\mathbf  x)$ can be viewed as a faithful summary of the contribution of each component of $\mathbf  x$ to the output $\mathbf  y$. For standard piece-wise linear models without bias, such a $\mathbf  W(\mathbf  x)$ can be obtained by computing the gradients with respect to input, as done in X-DNNs [20]. However, X-DNN explanations are not as human interpretable as compared to B-cos because of lack of alignment pressure. This is addressed by B-cos models [6], which with their B-cos transform obtain dynamic linearity differently, without piece-wise linearity, as discussed in Sec 3.2.1 in [6].
> > >
> > > **Normalized Weights:** Apologies for the overly short answer (due to space constraints) with respect to the question on weight normalisation, we gladly take this opportunity to elaborate in more detail. With our above answer, we tried to clarify on the motivation on using normalised weights as given in [6] — specifically, due to weight normalisation, a "B-cos neuron" (as defined by Eq. 3 in [6]) is bounded in its output strength and will produce its highest output if and only if w and x are co-linear, i.e., have maximal cosine-similarity; this property has been referred to as **weight-input alignment**. While this argument seems intuitive when considering B-cos layers in isolation, when combined with a subsequent normalisation layer (e.g., batch- or layer-normalisation), the model will become invariant to the usage of weight normalisation, as we describe in lines 170-183 in our submission. As a result, we find that B-cos networks without weight normalisation show similar properties as those reported in [6] and we thus agree with the reviewer that weight normalisation does indeed not seem necessary. We will revise our manuscript to make this point clearer.
> > >
> > > We unfortunately did not fully understand the question about 'Eq. 3' in the comment — we would be grateful if the reviewer could clarify so that we can fully resolve the reviewer's concerns. Specifically, $w$ does have a hat in the cosine term in Eq. 3 of [6].

---

### Author Rebuttal · Authors · 2024-08-06

We thank the reviewers for their detailed comments and constructive feedback. We are encouraged to find that the reviewers appreciate that our proposed approach allows us to convert pre-trained models to be interpretable whilst maintaining performance (Z7HS, DdYE). We are further encouraged to find that the reviewers find our work to constitute an interesting research direction (71N9, HC7z), our submission to be well structured (DdYE, JC7z), and to have a broad experimental evaluation (JC7z, DdYE).

While we are happy to find our submission to be generally very positively received, we of course also highly appreciate the various suggestions by the reviewers to improve our work. In the following, we address some of the concerns shared by multiple reviewers, with more detailed responses to be found in the respective rebuttal sections:
- **Repeat experiments multiple times (Z7HS, JC7z):** We now provide results over three runs corresponding to Tables 2, 3, and 4 in Tables 5, 6, and 7, respectively, in the attached PDF. From Tables 5-6, we find that the conclusions in Section 3.2.2 remain unchanged — the simplest approach of directly setting B=2 and removing the bias at the beginning performs as well as more involved strategies such as learning B or decaying bias. Also, from Table 7, we find that the results regarding classification and interpretability performance, as in Section 4.1, are supported by repeated experiments.
- **Localization scores vs. epochs (71N9, DdYE):** Thank you for the suggestion. We now report these results in Figure 6 in the attached PDF. We find both quantitatively (left) and qualitatively (right) that the localization improves very quickly. For example, for ResNet-18, the localization improves from 21.5 to 82.0 after just one epoch of fine-tuning, which is close to 86.9 achieved by the B-cos model trained from scratch. On the other hand, modifications to the model architecture at the beginning of the B-cosification process lead to an initial drop in accuracy, from 69.8 for the standard model to 59.9 after one epoch. However, we recover the accuracy much quicker than training from scratch, with the B-cosified model needing just 29 epochs to reach the accuracy of a B-cos model trained from scratch for 90 epochs.
- Finally, we will carefully proofread the final version of the manuscript and incorporate the valuable suggestions regarding the clarity of the writing and experimental evaluation given by the reviewers.

---

### Decision · Program_Chairs · 2024-09-25

**Decision:**

Accept (poster)

**Comment:**

The paper proposes a method to convert a regular network into a more interpretable network (a B-cos network). This addresses a major limitation in B-cos networks that previously had to be trained from scratch. Interpretability is a challenging and important area, and the results in this paper are compelling. The reviewers unanimously vote for acceptance, and I agree. Congratulations to the authors!